# Quaternion Algebra on 4D Superfluid Quantum Space-Time: Can Dark Matter Be a Manifestation of the Superfluid Ether?

**Valeriy Sbitnev** [1,2]

1   St. Petersburg B. P. Konstantinov Nuclear Physics Institute, NRC Kurchatov Institute, 188300 Gatchina, Russia; valery.sbitnev@gmail.com or sbitnev_vi@pnpi.nrcki.ru; Tel.: +7-8137137944

2   Department of Electrical Engineering and Computer Sciences, University of California, Berkeley, CA 94720, USA

**Abstract:** Quaternions are a natural framework of 4D space-time, where the unit element relates to time, and three others relate to 3D space. We define a quaternion set of differential torsion operators (shifts with rotations) that act to the energy-momentum tensor written on the same quaternion basis. It results in the equations of gravity-torsion (gravitomagnetic) fields that are similar to Maxwell's equations. These equations are parent equations, generating the following equations: (a) equations of the transverse gravity-torsion waves; (b) the vorticity equation describing vortices orbital speed of which grows monotonically in the vortex core but far from it, it goes to a permanent level; (c) the modified Navier–Stokes equation leading to the Schrödinger equation in the nonrelativistic limit and to the Dirac equation in the relativistic limit. The Ginsburg–Landau theory of superfluidity resulting from the Schrödinger equation shows the emergence of coupled proton-antiproton pairs forming the Bose–Einstein condensate. In the final part of the article, we describe Samokhvalov's experiment with rotating nonelectric, nonferromagnetic massive disks in a vacuum. It demonstrates an unknown force transferring the rotational moment from the driving disk to a driven one. It can be a manifestation of the dark matter. For studying this phenomenon, we propose a neutron interference experiment that is like the Aharonov–Bohm one.

**Keywords:** gravitomagnetism; superfluid; dark matter; quark; hexaquark; Bose–Einstein condensate; quantum ether; quaternion; vorticity



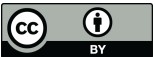

## 1. Introduction

The observed universe evolves in a four-dimensional pseudo-Euclidean Minkowski space-time, with a metric tensor, $\eta_{\mu,\nu}$, having a space-like signature $(-,+,+,+)$, $\mu$ and $\nu$ run from 0 to 3. Where three coordinates are Cartesian coordinates of three-dimensional Euclidean space. Furthermore, the fourth zeroth-coordinate is time multiplied by the speed of light $c$, so its dimension is the length also. The relativistic hydrodynamics starts from a description of the energy-momentum tensor [1]:

$$T^{\mu,\nu} = (\epsilon + p)u^{\mu}u^{\nu} + p\eta^{\mu,\nu}. \tag{1}$$

Here $\epsilon$ and $p$ are the energy and momentum functions per unit volume and $u^{\mu} = \gamma(c, \vec{v})$ is the four-velocity meeting to the condition $u_{\mu}u^{\mu} = -c^2$, where $\gamma = (1 - v^2/c^2)^{-1/2}$. This theoretical discipline is a very productive instrument for describing the dynamics of matter from scales as small as those of colliding elementary particles up to the largest scales in the universe [2–5].

The cosmological principle states that the spatial distribution of matter in the universe is homogeneous and isotropic when viewed on a large enough scale. As was supported recently by Planck collaboration, our universe has zeroth curvature [6]. The spatial curvature is found to be $|\Omega_K| < 0.005$. We live in a flat universe shrouded in the ether that fills the space everywhere densely. Let us imagine, in the first approximation, that the

surrounding medium, the ether, behaves itself as a fluid. In the case of the perfect fluid the energy-momentum tensor (1) has a diagonal form $T^{0,0} = \epsilon_0$, $T^{1,1} = p_x$, $T^{2,2} = p_y$, $T^{1,1} = p_z$. When the fluid is at rest all nondiagonal terms of the energy-momentum tensor are zero [7].

One should note that, based on the results of Michelson and Morley interferometry, the ether, recognized since ancient times as a space filler, does not exist. All the surrounding space is a cosmic vacuum in which the visible universe evolves. The cosmic vacuum is not empty, however, but on small scales, there are quantum zero-point fluctuations, expressed through the uncertainty principle:

$$\Delta \vec{p} \cdot \Delta \vec{r} \geq \frac{\hbar}{2}, \qquad \Delta \epsilon \cdot \Delta t \geq \frac{\hbar}{2}. \tag{2}$$

The first inequality characterizes the impossibility of the instant registration of both the particle momentum and its position. The second one remarks the impossibility of the instant registration of time and particle energy. Taking into account that $\Delta \vec{p} = m\Delta\vec{v} = m\Delta\vec{r}/\Delta t$ we get from the above inequalities the diffusion length of particle migration

$$\Delta r = \sqrt{D\Delta t} \tag{3}$$

where

$$D = \frac{\hbar}{2m} \tag{4}$$

is the diffusion coefficient [8]. Nelson hypothesizes that a free particle in space, or, say, the ether, undergoes Brownian motion with a diffusion coefficient proportional to the reduced Planck constant, $\hbar$, and inverse proportional to the inertial mass $m$. [9,10]. It undergoes random collisions with unknown virtual particles filling ether, scattering on the diffusion length $\Delta\vec{r}$. One may imagine that motion of the virtual particles has a weak dispersion around a certain average. Further, we can attract the 4D Fick's law for getting the 4D quantum potential $Q$ [1] from the energy-momentum tensor (1). As a result, we have

$$Q = -2mD^2 \frac{\sqrt{\rho}}{\sqrt{\rho}}. \tag{5}$$

Here $\rho$ is the probability density of detecting particle within the unit volume under observation.

From the above, it follows that the quantum particle undergoes the energy dissipation at motion through the ether. By common sense, with time, it should lose the energy up to the zero-point energy fluctuations. To describe this loss, we add to the energy-momentum tensor the viscous stress tensor $\Pi^{\mu,\nu}$. As a result, the energy-momentum tensor adopts the following view

$$T^{\mu,\nu} = (\epsilon + p)u^\mu u^\nu + p\eta^{\mu,\nu} + \Pi^{\mu,\nu}. \tag{6}$$

In the general case, we represent it in the following view [4]

$$\Pi^{\mu,\nu} = \mu c(\partial^\mu u^\nu + \partial^\nu u^\mu) + c\left(\zeta - \frac{2}{3}\mu\right)\partial^\mu u_\mu \eta^{\mu,\nu} \tag{7}$$

This viscosity tensor describes both bulk and shear viscous dissipations induced by pressure gradients evolving in the medium. These two forces (the pressure gradients and viscous frictions in the medium resulting in dissipative losses) are inner ones that arise as a counteraction to external forces applied to a unit of volume of the medium from outside.

One can guess that outer space is a complex medium where dissipative forces play a crucial role. This remark follows, in particular, from measured a CMB temperature equal about to 2.726 K [11]. There is reason to believe that this medium is a dilatant fluid [12]. It means that its viscosity increases when speed tends to the speed of light. What a surprise it was when it turned out that the entire observable universe, in addition to everything

else, is everywhere densely surrounded by a dark, incomprehensible substance that is not observable in any way, except for its pronounced gravitational effect.

The dark matter hypothesis appeared in the 1930s when observing the rotation of galaxies. Fritz Zwicky discovered in 1933 [13] a large spread of radial velocities of eight galaxies in the Coma cluster (the constellation of Veronica's hair). The apparent speed of rotation of stars in galaxies did not correspond to their apparent mass. Applying the virial theorem, he concluded that the stability of the cluster of its total mass must be 400 times greater than that of its stars. He concluded that the so-called "hidden mass" occupies most of the Universe. It is an invisible substance that manifests itself by interacting with the visible through the forces of gravity only. The mass of this substance is many times greater than that of all observed objects.

While studying the rotation curves of galaxies, Vera Rubin found discrepancies between the predicted circular motion of galaxies and the observed motion [14,15]. This fact, which became known as the "galaxy rotation problem" [16], became one of the main pieces of evidence for the existence of dark matter.

According to the observations of the Planck space Observatory, published in 2013 [17], we know that dark matter accounts for about 27% of the total mass of the Universe, 68% is accounted for by dark energy. Furthermore, only the remaining 5% is occupied by visible, baryon matter. At the same time, the lion's share from these 5% falls on free hydrogen (about 74%) and helium (about 24%). The remainder, about 2%, is distributed by all other elements of the periodic table of elements.

In the beginning, it was assumed that galaxies have a lot of interstellar gas, wandering planets, and burnt-out brown dwarfs, that is, an ordinary matter that does not glow, and we do not see it. However, calculations and observations have shown the fallacy of this assumption. If there were so much interstellar gas in galaxies, then the processes of star formation in them would be much more intense than we see.

Various variants of the dark matter composition were proposed, up to very exotic ones. However, attempts to find any signatures of dark matter have not yet been successful. There are even works that deny the existence of dark matter [1,18–21]. In any case, it makes sense to follow the principle of Occam's razor and not involve unnecessary entities if it is possible to explain the unobserved phenomenon by minimal means. In particular, as we noted above, hydrogen shows the highest percent of its content in the universe. There is a plausible hypothesis that "cold dark matter" represents slow interstellar and intergalactic neutral atomic hydrogen in its main lower 1st ground state [22,23].

There is a reason to believe that a candidate onto the dark matter and energy can be Bose–Einstein condensate (BEC) [24–30]. In favor of this supposition, the following fact witnesses—the superconducting medium expels the force lines of an external magnetic field outside its volume. It is the Meissner effect [31]. Due to this effect, such a medium becomes invisible to the electromagnetic fields. BEC representing a superfluid quantum medium [32–35] is an updated version of the ether [36–39]. Now it is the superfluid quantum ether [34,40]. The space filled everywhere densely by the superfluid quantum medium named ether represents the superfluid quantum space-time [41,42].

Which can mathematical apparatus be for dealing with such a superfluid quantum space-time? First, note that all things in the universe possess inertial masses, and they are in constant rotation. Everything in this world revolves, beginning from particles and up to galaxies. Rodger Penrose proposed a new type of algebra for Minkowski space-time, in terms of which it is possible to express any conformally covariant or Poincar's covariant operation [43,44]. He argues that such rescalings naturally lead to torsion in space-time. Which is consistent with the local torsion of the Einstein–Cartan–Sciama–Kibble [45–47].

Following this course, we remark first that 4D space-time can be represented mathematically by introducing four orthogonal units $\mathbf{1}, \mathbf{i}, \mathbf{j}, \mathbf{k}$, first defined by Sir William Rowan Hamilton and named quaternions [48]. James Clerk Maxwell, the discoverer of electromagnetic theory, wrote about the discovery of the quaternion calculus [49]: *"The invention of the calculus of quaternions is a step towards the knowledge of quantities related to space which can only*

*be compared, for its importance, with the invention of triple coordinates by Descartes. The ideas of this calculus, as distinguished from its operations and symbols, are fitted to be of the greatest use in all parts of science."* The quaternions are hypercomplex numbers [50–58] forming a vector space of dimension four over the field of real numbers:

$$\vec{q} = a\mathbf{1} + b\mathbf{i} + c\mathbf{j} + d\mathbf{k}. \tag{8}$$

A feature of quaternions is that multiplication of two quaternions is noncommutative: $\mathbf{i} \cdot \mathbf{j} = \mathbf{k}$ while $\mathbf{j} \cdot \mathbf{i} = -\mathbf{k}$. Multiplication of quaternions is expressed in the following table:

$$
\begin{array}{c}
\phantom{\mathbf{1}}\;\;\mathbf{1}\quad\mathbf{i}\quad\mathbf{j}\quad\mathbf{k} \\
\begin{array}{c}\mathbf{1}\\\mathbf{i}\\\mathbf{j}\\\mathbf{k}\end{array}
\begin{pmatrix}
\mathbf{1} & \mathbf{i} & \mathbf{j} & \mathbf{k} \\
\mathbf{i} & -\mathbf{1} & \mathbf{k} & -\mathbf{j} \\
\mathbf{j} & -\mathbf{k} & -\mathbf{1} & \mathbf{i} \\
\mathbf{k} & \mathbf{j} & -\mathbf{i} & -\mathbf{1}
\end{pmatrix}.
\end{array}
\tag{9}
$$

To move deeper into the understanding of the superfluid quantum space-time, we invoke quaternion algebra. It permits analyzing the different degrees of freedom of our 4D space-time, both translations and torsions. Section 2 deals with this algebra and its possibilities. Section 3 opens the gravitomagnetic fields. The following section, Section 4, contains four subsections extracting the wave equations, the vorticity equation, and the Schrödinger and Dirac equations. In this section, we consider the Ginzburg–Landau theory of superfluidity. It deals with paired protons and antiprotons that are buffered by the electron-positron pairs. They form BEC. Further, it leads to the consideration of quark models of baryon particles, hexaquarks underlying BEC. In Section 5, we describe Samochvalov's experiment with rotating nonferromagnetic electrically neutral massive disks in a deep vacuum. There is reason to believe that this experiment reveals the mystery of dark matter. We propose a neutron interference experiment similar to the Aharonov–Bohm experiment for observing this rotating mass-dynamical effect. Section 6 proposes the concluding remarks.

## 2. Quaternion Algebra and the Energy-Momentum Tensor of Gravitomagnetic Field

There is one-to-one correspondence between four quaternions $\mathbf{1}$, $\mathbf{i}$, $\mathbf{j}$, $\mathbf{k}$ and four $2 \times 2$ matrices $\sigma_0$, $\sigma_x$, $\sigma_y$, $\sigma_z$, of the group SU(2):

$$\sigma_0 = \begin{pmatrix} 1 & 0 \\ 0 & 1 \end{pmatrix} \quad \sigma_x = \begin{pmatrix} 0 & 1 \\ 1 & 0 \end{pmatrix} \quad \sigma_y = \begin{pmatrix} 0 & -\mathbf{i} \\ \mathbf{i} & 0 \end{pmatrix} \quad \sigma_z = \begin{pmatrix} 1 & 0 \\ 0 & -1 \end{pmatrix}. \tag{10}$$

Rodger Penrose uses these matrices for applying the twistor methods in Space-Time Geometry. Note that a single matrix, $\sigma_y$, contains the imaginary unit. Because of this, at matrices multiplications, the group SU(2) poses the complex $2 \times 2$ matrices. This is not good, since it leads to complicated algebra dealing with complex numbers. However, there is one-to-one correspondence between the four $2 \times 2$ matrices $\sigma_0$, $\sigma_x$, $\sigma_y$, $\sigma_z$ and $4 \times 4$ matrices components of which are all real numbers. These matrices were successfully applied at studying the spin neutron resonance and demonstrated perfect abilities [59–61]. These matrices form the quaternion group $\mathbb{H}^2$ [52–55]. One can consider the quaternion algebra on 4D space-time as a lateral branch of Penrose's twister program.

Summarize we can say that there is one-to-one correspondence between four quaternions, $\mathbf{1}$, $\mathbf{i}$, $\mathbf{j}$, $\mathbf{k}$, four $2 \times 2$ matrices, $\sigma_0$, $\sigma_x$, $\sigma_y$, $\sigma_z$, of the group SU(2), and four $4 \times 4$ matrices of the quaternion group $\mathbb{H}^2$. The latter matrices look as follows:

$$\eta_0 = \begin{pmatrix} 1 & 0 & 0 & 0 \\ 0 & 1 & 0 & 0 \\ 0 & 0 & 1 & 0 \\ 0 & 0 & 0 & 1 \end{pmatrix}, \tag{11}$$

$$\eta_x = \begin{pmatrix} 0 & -1 & 0 & 0 \\ 1 & 0 & 0 & 0 \\ 0 & 0 & 0 & 1 \\ 0 & 0 & -1 & 0 \end{pmatrix}, \quad \eta_y = \begin{pmatrix} 0 & 0 & -1 & 0 \\ 0 & 0 & 0 & -1 \\ 1 & 0 & 0 & 0 \\ 0 & 1 & 0 & 0 \end{pmatrix}, \quad \eta_z = \begin{pmatrix} 0 & 0 & 0 & -1 \\ 0 & 0 & 1 & 0 \\ 0 & -1 & 0 & 0 \\ 1 & 0 & 0 & 0 \end{pmatrix}. \tag{12}$$

Matrices (11) and (12) submit to the rules of multiplication

$$\begin{array}{c|cccc} & \eta_0 & \eta_x & \eta_y & \eta_z \\ \hline \eta_0 & \eta_0 & \eta_x & \eta_y & \eta_z \\ \eta_x & \eta_x & -\eta_0 & \eta_z & -\eta_y \\ \eta_y & \eta_y & -\eta_z & -\eta_0 & \eta_x \\ \eta_z & \eta_z & \eta_y & -\eta_x & -\eta_0 \end{array}. \tag{13}$$

Compare it with Equation (9).

So, instead of using the spinor complex matrices of the group SU(2) applied for describing 4D space-time [46,47] we will go to quaternion matrices (11) of the group $\mathbb{H}^2$ with real numbers. Therefore, the quaternion algebra on 4D space-time can be viewed to be a lateral branch of Penrose's twister program. A general quaternion matrix representation has the following form [42,62,63]:

$$\mathcal{D}(\vec{u}) = u_0\eta_0 + u_x\eta_x + u_y\eta_y + u_z\eta_z \tag{14}$$

where $u_0$, $u_x$, $u_y$, $u_z$ are arbitrary variables or operators.

Running ahead we note that we will attach the imaginary unit to the time axis, while the spatial coordinates are real variables. Such an assignment makes sense. The cones of the past and future represent sets of imaginary events: the part has already occurred but others can yet occur under certain circumstances. First, we define differential operators having the following representation

$$\begin{aligned} \mathcal{D} &= \mathbf{i}c^{-1}\partial_t\eta_0 + \partial_x\eta_x + \partial_y\eta_y + \partial_z\eta_z, \\ \mathcal{D}^{\mathbf{T}} &= \mathbf{i}c^{-1}\partial_t\eta_0^{\mathbf{T}} + \partial_x\eta_x^{\mathbf{T}} + \partial_y\eta_y^{\mathbf{T}} + \partial_z\eta_z^{\mathbf{T}} \\ &= \mathbf{i}c^{-1}\partial_t\eta_0 - \partial_x\eta_x - \partial_y\eta_y - \partial_z\eta_z. \end{aligned} \tag{15}$$

Here $\partial_t = \partial/\partial t$, $\partial_x = \partial/\partial x$, etc., $c$ is the speed of light, and sign $\mathbf{T}$ means the transposition. These operators realize torsion shifts in time—translations with rotations. Note that the differential time operator contains the imaginary unit $\mathbf{i}$. Time is a special variable fixing moment of the change of spatial patterns that disappear in the past but have not yet emerged from the future. Zeno's aporias emphasize this particular feature of time—"there is only an instant between the past and the future, and this instant is called a life"©. The d'Alembertian or the wave operator for the case of the negative metric signature $\{-1, +1, +1, +1\}$ reads

$$\mathcal{D}^{\mathbf{T}}\mathcal{D} = \mathcal{D}\mathcal{D}^{\mathbf{T}} = (-c^{-2}\partial_t^2 + \partial_x^2 + \partial_y^2 + \partial_z^2)\eta_0 = \square. \tag{16}$$

Now we come to the determination of the matter and its motion in our 3D space. Let $\epsilon_0$ be its energy density, while $p_{x,0}$, $p_{y,0}$, $p_{z,0}$ be components of the momentum density. The latter multiplied by the speed of light, $c$, is adopted as the renormalized momentum density

$$\vec{p}_0 \leftarrow \vec{p}_0 \cdot c = \gamma(v)\rho_M\vec{v} \cdot c \approx \rho_M\vec{v}\cdot c. \tag{17}$$

In turn, the relativistic energy density, $\epsilon_0$, of what we call a particle, can be expressed in terms of its momentum density by the following expression

$$\epsilon_0 = \sqrt{\rho_M^2 c^4 + p_0^2} = \gamma(v)\rho_M c^2 \approx \rho_M c^2 + \rho_M\frac{v^2}{2}. \tag{18}$$

Here $\gamma(v) = (1 - v^2/c^2)^{-1/2}$ is the Lorentz factor. Nonrelativistic limit is $\gamma(v) \to 1$. These limits are written on the right hand in Equations (17) and (18). Note that these formulas are written under the assumption of relativistic transport fluxes in an ideal fluid [3].

Note that the values $\epsilon_0$ and $\vec{p}_0$ have equal dimensionality—the dimension of the energy density due to extra multiplying $\vec{p}_0$ by the speed of light $c$ [1]. The mass density, $\rho_M$, reads

$$\rho_M = \frac{M}{\Delta V} = m\frac{N}{\Delta V} = m\rho. \tag{19}$$

Here $\Delta V$ is the unit volume of the fluid under consideration. $M = mN$ is the total mass of the fluid medium that characterizes its inertial properties. $N$ is the number of carriers of the elementary mass $m$, and $\rho$ is the density of these simple carriers. The function $\rho$ plays an important role in the quantum realm. It determines the probability density of detecting particles within the volume $\Delta V$ at the moment $t$. In particular, this function enters to the definition of the quantum potential (5).

For the sake of completeness of the picture, we add to the momentum and energy densities the electromagnetic scalar and vector potentials [64]:

$$\epsilon = \epsilon_0 - \rho_e\Phi = \rho(m\gamma(v)c^2 - e\Phi). \qquad \vec{p} = \vec{p}_0 - \rho_e c\vec{A} = c\rho(m\gamma(v)\vec{v} - e\vec{A}), \tag{20}$$

Here $\rho_e = e\rho$ is the charge density per the unit volume, $\Delta V$. That is, each carrier of the elementary mass $m$ is also the carrier of the elementary charge $e$. We use SI units. Therefore, the vector potential $c\vec{A}$ is multiplied by the speed of light. Since $|\vec{A}|$ has the dimension $V \cdot s \cdot m^{-1}$ and $\Phi$ has the dimension of V, both variables, $\rho_e\Phi$ and $\rho_e c\vec{A}$, have the dimension of pressure, [Energy/m$^3$].

Now for performing the gauge transformation, we introduce two scalar fields. One scalar field, $\psi$, relates to the gauge transformation of the massive medium, and other fields, $\varphi$, deal with that of the massless medium, the electromagnetic field (EM field), carriers of which are EM quanta with zero mass. We rewrite Equation (20) as follows

$$\begin{aligned} \epsilon &:= (\epsilon_0 - \rho_e\Phi) - c^{-1}\partial_t(\psi - \varphi), \\ \vec{p} &:= (\vec{p}_0 - \rho_e c\vec{A}) + \nabla(\psi - \varphi), \end{aligned} \tag{21}$$

Here $\phi = \psi - \varphi$ is an arbitrary scalar field, having dimensionality of Energy $\times$ Length$^{-2}$. In the light of this remark, we write the energy-momentum density tensor with added the extra term $\mathcal{D}^\mathbf{T}\phi$:

$$T = \mathbf{i}(\epsilon_0 - \rho_e\Phi)\eta_0 + (p_{x,0} - \rho_e cA_x)\eta_x + (p_{y,0} - \rho_e cA_y)\eta_y + (p_{z,0} - \rho_e cA_z)\eta_z - \mathcal{D}^\mathbf{T}\phi. \tag{22}$$

The term $\mathcal{D}^\mathbf{T}\phi = \mathcal{D}^\mathbf{T}(\psi - \varphi)$ returns the expression $\mathbf{i}c^{-1}\partial_t\eta_0\phi - (\nabla \cdot \vec{\eta})\phi$ that comes from the arbitrary scalar field $\phi$ added in Equation (21). It should be noticed that in contrast to the customary defined energy-momentum tensor (1) here $\epsilon$, $p_x$, $p_y$, $p_z$ are multipliers at the matrices $\eta_0$, $\eta_x$, $\eta_y$, $\eta_z$. The components $p_x$, $p_y$, $p_z$ do not lie on the diagonal. Instead they form a quaternion triad reproducing the curl operation.

Further, we define the Lorentz gauge condition

$$\frac{1}{4}\text{trace } \mathcal{D}T = -\frac{1}{c}\partial_t\epsilon - \partial_x p_x - \partial_y p_y - \partial_z p_z - \Box\phi = 0. \tag{23}$$

The term $\phi = \mathcal{D}^\mathbf{T}\mathcal{D}\phi$ represents a wave equation of the scalar field $\phi = \psi - \varphi$. One can see that the presence of the EM field shifts the scalar field $\psi$ of the medium loaded by mass.

Let us now rewrite out the product $\mathcal{D} \cdot T$, in details

$$
\begin{aligned}
\mathcal{D} \cdot T = & \left( -\frac{1}{c}\frac{\partial \epsilon}{\partial t} - \frac{\partial p_x}{\partial x} - \frac{\partial p_y}{\partial y} - \frac{\partial p_z}{\partial z} - \phi \right) \eta_0 \\
& + \left( \mathbf{i}\left( \frac{\partial \epsilon}{\partial x} + \frac{\partial p_x}{c \partial t} \right) - \left( \frac{\partial p_z}{\partial y} - \frac{\partial p_y}{\partial z} \right) \right) \eta_x \\
& + \left( \mathbf{i}\left( \frac{\partial \epsilon}{\partial y} + \frac{\partial p_y}{c \partial t} \right) - \left( \frac{\partial p_x}{\partial z} - \frac{\partial p_z}{\partial x} \right) \right) \eta_y \\
& + \left( \mathbf{i}\left( \frac{\partial \epsilon}{\partial z} + \frac{\partial p_z}{c \partial t} \right) - \left( \frac{\partial p_y}{\partial x} - \frac{\partial p_x}{\partial y} \right) \right) \eta_z
\end{aligned}
\tag{24}
$$

One can see that the multiplier of the unit quaternion $\eta_0$ represents the Lorentz gauge when it is zero. Only the expressions of the quaternions $\eta_x$, $\eta_y$, $\eta_z$ remain, which do not contain terms with $\phi$. As a result, we may rewrite this equation in the following manner

$$
\mathbb{F}_{\mathcal{E B}} = \mathcal{D} \cdot T = \begin{pmatrix} 0 & \mathcal{B}_x - \mathbf{i}\mathcal{E}_x & \mathcal{B}_y - \mathbf{i}\mathcal{E}_y & \mathcal{B}_z - \mathbf{i}\mathcal{E}_z \\ -\mathcal{B}_x + \mathbf{i}\mathcal{E}_x & 0 & -\mathcal{B}_z + \mathbf{i}\mathcal{E}_z & \mathcal{B}_y - \mathbf{i}\mathcal{E}_y \\ -\mathcal{B}_y + \mathbf{i}\mathcal{E}_y & \mathcal{B}_z - \mathbf{i}\mathcal{E}_z & 0 & -\mathcal{B}_x + \mathbf{i}\mathcal{E}_x \\ -\mathcal{B}_z + \mathbf{i}\mathcal{E}_z & -\mathcal{B}_y + \mathbf{i}\mathcal{E}_y & \mathcal{B}_x - \mathbf{i}\mathcal{E}_x & 0 \end{pmatrix}
\tag{25}
$$

This form is the gravitomagnetic tensor represented in the quaternion basis. In this basis, the electromagnetic tensor shows the same form [63]. The complex force field, the gravitomagnetic field $\vec{\mathcal{B}} - \mathbf{i}\vec{\mathcal{E}}$, as follows from (24), reads:

$$
\vec{\mathcal{B}} = [\nabla \times \vec{p}],
\tag{26}
$$

$$
\vec{\mathcal{E}} = \frac{\partial \vec{p}}{c \partial t} + \nabla \epsilon.
\tag{27}
$$

Since we introduce the electromagnetic field added by Equation (20) this field contains both the gravito-torsion field, $\vec{\Omega} - \mathbf{i}\vec{\Xi}$, and the electromagnetic field, $\rho_e(c\vec{B} + \mathbf{i}\vec{E})$:

$$
\vec{\mathcal{B}} - \mathbf{i}\vec{\mathcal{E}} = (\vec{\Omega} - \mathbf{i}\vec{\Xi}) - \rho_e(c\vec{B} + \mathbf{i}\vec{E}).
\tag{28}
$$

So, as follows Equation (20) the magnetic and electric fields subject to the following equations

$$
\vec{B} = [\nabla \times \vec{A}],
\tag{29}
$$

$$
\vec{E} = -\frac{\partial \vec{A}}{c \partial t} - \nabla \Phi.
\tag{30}
$$

Take attention that Equations (27) and (30) have opposite signs, while Equations (26) and (29) have equal signs. This can mean that the gravitomagnetic and electromagnetic equations, as we will see further, can have different signs. Note in this place that the gravitomagnetic equations of Oliver Heaviside [65], which he wrote out by analogy with Maxwell's equations, have equal signs with the latter equations. We will use the term "gravitomagnetism" first introduced by Oliver Heaviside since this term is well-established in the scientific literature [66–69].

## 2.1. Lorentz Transformation of the Gravitomagnetic Field Tensor

The Lorentz transformation is that of coordinates between two frames of reference that move at constant velocity relative to each other. The Lorentz transformation of the

4-dimensional space-time spanned on the quaternion basis, $(\eta_0, \eta_x, \eta_y, \eta_z)$, is carried out by matrices of the view [63]

$$
\begin{aligned}
L &= v_0\eta_0 + v_x\eta_x + v_y\eta_y + v_z\eta_z, \\
L^{\mathbf{T}} &= v_0\eta_0 - v_x\eta_x - v_y\eta_y - v_z\eta_z.
\end{aligned}
\tag{31}
$$

We require that there is always the equality $L \cdot L^{\mathbf{T}} = L^{\mathbf{T}} \cdot L = \eta_0$. From here it follows that the parameters $v_0, v_x, v_y, v_z$ submit to the following constraint

$$
v_0^2 + v_x^2 + v_y^2 + v_z^2 = 1.
\tag{32}
$$

We can write two types of transformations. They are rotation of the coordinate system in three-dimensional space and the transition to the moving coordinate system (boost) [46,47]:

- transformations of the coordinate system in the Minkowski space associated with rotation about the axis $\vec{m} = (m_x, m_y, m_z)$ on an angle $\alpha$ one can realize by the following trigonometric functions

$$
v_0 = \cos\left(\frac{\alpha}{2}\right), \qquad \vec{v} = \vec{m} \cdot \sin\left(\frac{\alpha}{2}\right);
\tag{33}
$$

- transformations of the coordinate system in the Minkowski space associated with the shift in the direction of the axis $\vec{m} = (m_x, m_y, m_z)$ on a boost $\beta = v/c$ ($v$ is a speed of the moving laboratory coordinate system) one can realize by the following hyperbolic functions

$$
v_0 = \cosh\left(\frac{\beta}{2}\right), \qquad \vec{v} = \vec{m} \cdot \sinh\left(\frac{\beta}{2}\right);
\tag{34}
$$

During the transition to the new coordinate system, the gravitomagnetic field tensor $\mathbb{F}_{\mathcal{EB}}$ experiences the transformation

$$
\mathbb{F}'_{\mathcal{EB}} = L \cdot \mathbb{F}_{\mathcal{EB}} \cdot L^{\mathbf{T}}.
\tag{35}
$$

After a series of computations, we come to the following transformation of the complex gravitomagnetic field $\vec{\mathcal{F}} = -\vec{\mathcal{B}} + \mathbf{i}\vec{\mathcal{E}}$:

$$
\vec{\mathcal{F}}' =
\begin{pmatrix}
-\mathcal{B}'_x + \mathbf{i}\mathcal{E}'_x \\
-\mathcal{B}'_y + \mathbf{i}\mathcal{E}'_y \\
-\mathcal{B}'_z + \mathbf{i}\mathcal{E}'_z
\end{pmatrix}
= R(\vec{v})
\begin{pmatrix}
-\mathcal{B}_x + \mathbf{i}\mathcal{E}_x \\
-\mathcal{B}_y + \mathbf{i}\mathcal{E}_y \\
-\mathcal{B}_z + \mathbf{i}\mathcal{E}_z
\end{pmatrix}
= R(\vec{v})\vec{\mathcal{F}}.
\tag{36}
$$

The $3 \times 3$ matrix $R(\vec{v}) = R(v_0, v_x, v_y, v_z)$ resulting from the computations of Equation (35) looks as follows:

$$
R(v_0, v_x, v_y, v_z) =
\begin{pmatrix}
2v_x^2 - 1 & 2(v_x v_y + v_0 v_z) & 2(v_x v_z - v_0 v_y) \\
2(v_y v_x - v_0 v_z) & 2v_y^2 - 1 & 2(v_y v_z + v_0 v_x) \\
2(v_z v_x + v_0 v_y) & 2(v_z v_y - v_0 v_x) & 2v_z^2 - 1
\end{pmatrix}
\tag{37}
$$

By writing out in clear view the transformation (36) we get the following expressions

- when rotating the coordinate system about the axis $\vec{m}$ on the angle $\alpha$ it reads

$$
\vec{\mathcal{F}}' = \vec{\mathcal{F}} \cdot \cos(\alpha) + \vec{m}(\vec{m} \cdot \vec{\mathcal{F}}) \cdot (1 - \cos(\alpha)) - [\vec{m} \times \vec{\mathcal{F}}] \cdot \sin(\alpha).
\tag{38}
$$

One can see that the gravitomagnetic fields undergo rotations independently from each other;

- when boosting into the coordinate system moving with the velocity $\vec{v}$ along the direction $\vec{m} = \vec{v}/|v|$ concerning the initial coordinate system, the Lorentz transformation shifts by a value $\phi$ to be named the rapidity

$$\vec{\mathcal{F}}' = \vec{\mathcal{F}} \cdot \cosh(\phi) + \vec{m}(\vec{m} \cdot \vec{\mathcal{F}}) \cdot (1 - \cosh(\phi)) - \mathbf{i}[\vec{m} \times \vec{\mathcal{F}}] \cdot \sinh(\phi). \quad (39)$$

Note that $\cosh(\phi) = \gamma = 1/\sqrt{1 - \beta^2}$ is the Lorentz factor and $\sinh(\phi) = \beta\gamma$, here $\beta = v/c$ is the velocity coefficient into the $\vec{m}$-direction. By expressing this formula through the gravitomagnetic fields $\vec{\mathcal{E}}$ and $\vec{\mathcal{B}}$ we get

$$\vec{\mathcal{E}}' = \gamma\vec{\mathcal{E}} - (\gamma - 1)(\vec{\mathcal{E}}\cdot\vec{v})\vec{v}/v^2 + (\gamma/c)[\vec{v} \times \vec{\mathcal{B}}], \quad (40)$$
$$\vec{\mathcal{B}}' = \gamma\vec{\mathcal{B}} - (\gamma - 1)(\vec{\mathcal{B}}\cdot\vec{v})\vec{v}/v^2 - (\gamma/c)[\vec{v} \times \vec{\mathcal{E}}]. \quad (41)$$

Since the gravitomagnetic field is the superposition of the gravity-torsion field and the electromagnetic field, see Equation (28), these formulas are valid for both electric, magnetic, gravitation, and torsion fields. As for the latter fields, these formulas show that at $v$ tending to the speed of light the torsion field transforms into the gravitation field, but the gravitation field transforms the torsion field.

### 2.2. Quadratic Forms of the Gravitomagnetic Field Tensor

The gravitomagnetic tensor permits the existence of two quadratic forms. The first quadratic form

$$\frac{1}{2}\mathbb{F}_{\mathcal{EB}}\mathbb{F}_{\mathcal{EB}}^{\dagger} = -W_0\eta_0 + \mathbf{i}W_x\eta_x + \mathbf{i}W_y\eta_y + \mathbf{i}W_z\eta_z \quad (42)$$

contains information about the density and flux of the gravitomagnetic energy. Here $\dagger$ is the sign of complex conjugation, $W_0 = (\mathcal{E}^2 + \mathcal{B}^2)/2$ is the energy density, and $\vec{W} = [\vec{\mathcal{E}} \times \vec{\mathcal{B}}]$ describes the energy flux in the direction perpendicular to the fields $\vec{\mathcal{E}}$ and $\vec{\mathcal{B}}$. Accurate to the divider $\mu_0$ (the vacuum permeability) vector $\vec{W}$ represents the Poynting vector.

The second quadratic form reads

$$\frac{1}{2}\mathbb{F}_{\mathcal{EB}}^{\mathbf{T}}\mathbb{F}_{\mathcal{EB}} = \frac{(\mathcal{B}^2 - \mathcal{E}^2)}{2}\eta_0 - \mathbf{i}(\vec{\mathcal{E}}\cdot\vec{\mathcal{B}})\eta_0. \quad (43)$$

It gives two invariants concerning the Lorentz transformations. The first invariant $I_1 = (\mathcal{B}^2 - \mathcal{E}^2)/2$ is the scalar and the second invariant $I_2 = (\vec{\mathcal{E}}\cdot\vec{\mathcal{B}})$ is the pseudoscalar.

## 3. The Gravitomagnetic Field Equations

The main equations of the gravitomagnetic fields result from the force density tensor, $\mathbb{F}_{\mathcal{EB}}$, after applying to it the differential operator $\mathcal{D}^{\mathbf{T}}$. For that reason, we need to define a 4D current of the force density that takes into account both gravitation and electromagnetic and other accompanying forces acting on the medium within the volume $\Delta V$ under consideration.

First, we need to define a density distribution term of all forces acting on the medium enclosed within the unit the volume $\Delta V$:

$$\wp = \frac{1}{4\pi}(\nabla \cdot \vec{F}). \quad (44)$$

The divisor $4\pi$ is to emphasize the commonality with Maxwell's EM equations. Here $\vec{F}$ is represented by the sum of all external and internal force densities, $f_1, f_2, \cdots$, acting on the fluid element under consideration [63]. In order not to deviate from the main content of this subsection, we will refrain from detailed considering the external and internal forces for the time being.

Let us define now 3D density current $\vec{\mathfrak{I}} = \vec{v}\,\wp$, where $\vec{v}$ is the velocity of the fluid element to be considered. Further, we introduce the 4D current density

$$\mathbb{J} = \mathbf{i}\,c\,\wp\,\eta_0 + \mathfrak{I}_x\eta_x + \mathfrak{I}_y\eta_y + \mathfrak{I}_z\eta_z. \quad (45)$$

The continuity equation, in this case, takes the following view

$$\frac{1}{4}\text{trace } \mathcal{D} \, \mathbb{J} - \partial_t \wp - \partial_x \Im_x - \partial_y \Im_y - \partial_z \Im_z = 0, \tag{46}$$

that can be rewritten in a more evident form

$$\frac{\partial \wp}{\partial t} + (\nabla \cdot \vec{\Im}) = 0. \tag{47}$$

In some sense, this equation is a manifestation of Newton's third law, the action-reaction law—all forces acting on a fluid element are in balance among themselves.

Let us now apply to the force density tensor (25) the differential operator $\mathcal{D}^{\mathbf{T}}$ and equate it to the 4D current density $\mathbb{J}$. We obtain a generating equation:

$$\mathcal{D}^{\mathbf{T}} \cdot \mathbb{F}_{\mathcal{E}\mathcal{B}} = \mathcal{D}^{\mathbf{T}} \mathcal{D} \cdot T = \Box T = \frac{4\pi}{c} \mathbb{J}. \tag{48}$$

By computing the product $\mathcal{D}^{\mathbf{T}} \cdot \mathbb{F}_{\mathcal{E}\mathcal{B}}$ we get the following set of terms collected as coefficients of the quaternion matrices $\eta_0, \eta_x, \eta_y, \eta_z$:

$$
\begin{aligned}
\mathcal{D}^{\mathbf{T}} \cdot \mathbb{F}_{\mathcal{E}\mathcal{B}} =\ & \{-(\partial_x \mathcal{B}_x + \partial_y \mathcal{B}_y + \partial_z \mathcal{B}_z) + \mathbf{i}(\partial_x \mathcal{E}_x + \partial_y \mathcal{E}_y + \partial_z \mathcal{E}_z)\}\eta_0 \\
& + \left\{\left(-\frac{1}{c}\partial_t \mathcal{E}_x - (\partial_y \mathcal{B}_z - \partial_z \mathcal{B}_y)\right) + \mathbf{i}\left(-\frac{1}{c}\partial_t \mathcal{B}_x + (\partial_y \mathcal{E}_z - \partial_z \mathcal{E}_y)\right)\right\}\eta_x \\
& + \left\{\left(-\frac{1}{c}\partial_t \mathcal{E}_y - (\partial_z \mathcal{B}_x - \partial_x \mathcal{B}_z)\right) + \mathbf{i}\left(-\frac{1}{c}\partial_t \mathcal{B}_y + (\partial_z \mathcal{E}_x - \partial_x \mathcal{E}_z)\right)\right\}\eta_y \\
& + \left\{\left(-\frac{1}{c}\partial_t \mathcal{E}_z - (\partial_x \mathcal{B}_y - \partial_y \mathcal{B}_x)\right) + \mathbf{i}\left(-\frac{1}{c}\partial_t \mathcal{B}_z + (\partial_x \mathcal{E}_y - \partial_y \mathcal{E}_x)\right)\right\}\eta_z \\
=\ & \frac{4\pi}{c}(\mathbf{i}\,c\,\wp\,\eta_0 + \Im_x \eta_x + \Im_y \eta_y + \Im_z \eta_z).
\end{aligned}
\tag{49}
$$

As seen from Equation (24) and this equation the operators $\mathcal{D}$ and $\mathcal{D}^{\mathbf{T}}$ generate curls shifting along in the space. Gathering together coefficients at the matrices $\eta_0, \eta_x, \eta_y, \eta_z$ represented in the right and left parts in Equation (49) we get the following pairs of the gravitomagnetic equations

$$(\nabla \cdot \vec{\mathcal{B}}) = 0, \tag{50}$$

$$[\nabla \times \vec{\mathcal{E}}] - \frac{1}{c}\frac{\partial}{\partial t}\vec{\mathcal{B}} = 0, \tag{51}$$

$$(\nabla \cdot \vec{\mathcal{E}}) = 4\pi \wp, \tag{52}$$

$$[\nabla \times \vec{\mathcal{B}}] + \frac{1}{c}\frac{\partial}{\partial t}\vec{\mathcal{E}} = -\frac{4\pi}{c}\vec{\Im}. \tag{53}$$

As noted in Equation (28) the gravitomagnetic fields $(\vec{\mathcal{E}}, \vec{\mathcal{B}})$ are represented by the superposition of the gravito-torsion fields $(\vec{\Xi}, \vec{\Omega})$ and the electromagnetic fields $(\vec{E}, \vec{B})$. Note that the latter fields are stronger then the former. However, in contrast to the latter, the former can not be shielded.

For the sake of comparison, it is instructive to give equations for the gravity-torsion and electromagnetic fields:

$$
\text{(a)}\ \left(
\begin{array}{l}
(\nabla \cdot \vec{\Omega}) = 0, \\[4pt]
[\nabla \times \vec{\Xi}] - \dfrac{1}{c}\dfrac{\partial}{\partial t}\vec{\Omega} = 0, \\[8pt]
(\nabla \cdot \vec{\Xi}) = 4\pi \wp, \\[4pt]
[\nabla \times \vec{\Omega}] + \dfrac{1}{c}\dfrac{\partial}{\partial t}\vec{\Xi} = -\dfrac{4\pi}{c}\vec{\Im}.
\end{array}
\right)
\qquad
\text{(b)}\ \left(
\begin{array}{l}
(\nabla \cdot \vec{B}) = 0, \\[4pt]
[\nabla \times \vec{E}] + \dfrac{1}{c}\dfrac{\partial}{\partial t}\vec{B} = 0, \\[8pt]
(\nabla \cdot \vec{E}) = 4\pi \rho_e, \\[4pt]
[\nabla \times \vec{B}] - \dfrac{1}{c}\dfrac{\partial}{\partial t}\vec{E} = \dfrac{4\pi}{c}\vec{j}_e.
\end{array}
\right)
\tag{54}
$$

In the second block of the equations $\rho_e = e\delta(\vec{r} - \vec{r}_0)$ and $\vec{j}_e = \vec{v}\rho_e$ are the charge and the 3D current densities, respectively. Here $\vec{v}$ is the velocity of the charge carriers within the fluid element under consideration.

Let us glance at the differences of signs in the equations written in the blocks (a) and (b). This discrepancy is shown qualitatively in Figure 1 taken from the book of Ignazio Ciufolini and John Archibald Wheeler entitled "gravitation and inertia" [70]. One can see that at equal orientations of the angular momentum $\vec{J}$ and the magnetic dipole momentum $\vec{m}$ the gravitomagnetic field $\vec{H}$ and the magnetic induction $\vec{B}$ have different orientations. One can note in this place that if we take for the valid picture as a magnetic field with the opposite direction, then the gravity-torsion and electromagnetic equations would be equivalent.

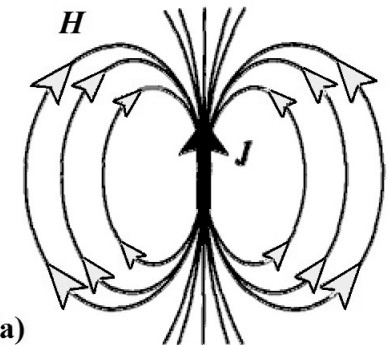 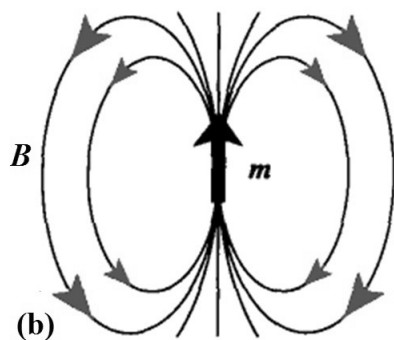

**(a)** **(b)**

**Figure 1.** The directionalities of gravitomagnetism and magnetism compared and contrasted [70]: (**a**) the gravitomagnetic field $\vec{H}$ in the weak field approximation, $\vec{J}$ is the angular momentum of the central body; (**b**) the magnetic induction $\vec{B}$ in the neighborhood of a magnetic dipole moment $\vec{m}$.

## 4. Equations Stemming from the Gravitomagnetic Equations

Three important equations can result from the gravitomagnetic equation. They are (a) the wave equations, (b) the vorticity equation, (c) the Schrödinger equation in the nonrelativistic limit, and the Dirac equation in the relativistic limit.

### 4.1. Wave Equations

The first result that one can get is wave equations. In this case, it is sufficient to apply to Equation (48) the differential operator $\mathcal{D}$:

$$\mathcal{D}\mathcal{D}^{\mathbf{T}} \cdot \mathbb{F}_{\mathcal{E}\mathcal{B}} = \frac{4\pi}{c}\mathcal{D}\mathbb{J} \tag{55}$$

and we immediately get [42]:

$$\frac{1}{c^2}\frac{\partial^2}{\partial t^2}\vec{\mathcal{B}} - \nabla^2\vec{\mathcal{B}} = -\frac{4\pi}{c}[\nabla \times \vec{\Im}], \tag{56}$$

$$\frac{1}{c^2}\frac{\partial^2}{\partial t^2}\vec{\mathcal{E}} - \nabla^2\vec{\mathcal{E}} = -4\pi\left(\nabla\wp + \frac{\partial}{c^2\partial t}\vec{\Im}\right). \tag{57}$$

Note that these wave equations describe both the electromagnetic waves and the gravity-torsion waves. The latter waves are weak, but they are not amenable to shielding,

In the light of the problems of astrophysics, the problem of generating these waves by a resonator having a certain temperature $T$ and whose size is comparable to the size of the visible universe is of interest. Its diameter is approximately equal to 28.5 gigaparsec [71] or about $8.8 \times 10^{26}$ m.

This problem is that of the black-body radiation. First, we may compute wave modes supported by this resonator that correspond to the energy levels of the harmonic oscillator

$$E_n = \hbar\omega\left(n + \frac{1}{2}\right), \qquad n = 0, 1, 2, \cdots . \tag{58}$$

The second step is to evaluate the thermal radiation of this resonator, which is given by Planck's formula for the distribution of energy in the radiation from the black body [72]. As a result, we come to the energy brightness of the black body

$$B_\omega(\omega, T) = \frac{\omega^2}{2\pi^2 c^2}\frac{\hbar\omega}{\exp\left\{\dfrac{\hbar\omega}{k_{\mathrm{B}}T}\right\} - 1}, \tag{59}$$

radiation of which occurs at the temperature $T$. In terms of parameters $h = 2\pi\hbar$ and $\nu = \omega/2\pi$ we obtain a formula for the cosmic microwave background (CMB) radiation as a function of the CMB radiation wavenumber, $k_{\mathrm{cmb}} = \nu/c$. Accurate to a scale factor $10^{20}$ it fits perfectly CMB spectrum measured by the FIRAS instrument on the COBE [73]. The temperature of measured CMB radiation is equal to $2.72548 \pm 0.00057$ K [11].

### 4.2. Vorticity Equations

The vorticity equation follows from the second gravitomagnetic equation, from Equation (51):

$$\frac{1}{c}\frac{\partial}{\partial t}\vec{\mathcal{B}} = [\nabla \times \vec{\mathcal{E}}]. \tag{60}$$

Here the vector field $\vec{\mathcal{B}}$ as follows from Equation (26) is $[\nabla \times \vec{p}]$. Taking into account Equation (20) we can rewrite the momentum in the nonrelativistic limit as follows

$$\vec{p} = c\rho_M(\vec{v} - (e/m)\vec{A}) = c\rho_M(\tilde{v}_{\,\mathrm{s}} + \tilde{v}_{\,\mathrm{o}}). \tag{61}$$

Here $\tilde{v}_{\,\mathrm{s}}$ is the irrotational velocity coming from gradient of the action function $S$ [74] and $\tilde{v}_{\,\mathrm{o}}$ is the orbital velocity [42] that is due to rotation of the carrier with mass $m$ and charge $e$. Note that $[\nabla \times \tilde{v}_{\,\mathrm{s}}] = 0$ while $[\nabla \times \tilde{v}_{\,\mathrm{o}}] = \vec{\omega}$ is not zero. The latter is called vorticity.

Now, let us glance at the right side of Equation (60). We note that the vector $\vec{\mathcal{E}}$ represents a force density acting on the unit volume of the fluid medium under consideration. In general, we can distinguish three force densities: (a) external forces acting on this element from the outside. These forces are conservative represented by the gradient of a scalar function, called the force function, which one can write as the negative gradient of a potential energy $U$; (b) internal forces that arise as a reaction at forces applied from the outside. These forces manifest themselves usually as pressure gradients. They lead to deformations of the medium (c) bulk viscosity force arising from the pressure gradients in the fluid are dissipative, which leads to heating of the medium. Looking ahead, we note that the pressure gradients manifest themselves as gradients from the quantum potential divided by the carrier density, $\rho$.

Note that the rotor of the gradient is zero. Therefore, the contribution of the first two forces mentioned in (a) and (b) vanishes. Only the third force can stay in Equation (60). Here for the sake of simplicity, we will consider the bulk viscosity $\mu\nabla^2\tilde{v}$ as a scalar term, acting uniformly over the entire volume of the medium under consideration [1]. The parameter $\mu$ relating the velocity shear to the stress is called the dynamical viscosity coefficient. Its dimension is [Newton·sec·m$^{-2}$].

Overwriting Equation (57) following the above-mentioned remarks, we get

$$\frac{\partial}{\partial t}\vec{\omega} = \nu(t)\nabla^2\vec{\omega}. \tag{62}$$

Here $\nu = \mu/\rho_M$ is the kinematic viscosity coefficient. Its dimension is [m$^2$·s$^{-1}$]. Here the viscosity coefficient is proposed to be dependent on time. The medium, filling all space everywhere densely, is superfluid [32,34,75,76]. It would seem that it makes sense to consider a medium with a zero viscosity coefficient, $\nu = 0$. However, this is a false path that sooner or later leads to the emergence of the singularities. We will do differently. We assume that the viscosity of the medium is zero on average over time, but its dispersion is not zero [74,77,78]:

$$\langle \nu(t) \rangle = 0_+, \qquad \langle \nu(t)\nu(0) \rangle > 0. \tag{63}$$

Let us look at the vortex tube in its cross-section, that oriented along the $z$-axis and with its center placed in the coordinate origin of the plane $(x, y)$, Figure 2. Equation (62), written down in the cross-section of the vortex, looks as follows

$$\frac{\partial \omega}{\partial t} = \nu(t)\left(\frac{\partial^2 \omega}{\partial r^2} + \frac{1}{r}\frac{\partial \omega}{\partial r}\right). \tag{64}$$

A general solution to this equation has the following view:

$$\omega(r, t) = \frac{\Gamma}{4\Sigma(\nu, t, \sigma)} \exp\left\{-\frac{r^2}{4\Sigma(\nu, t, \sigma)}\right\}, \tag{65}$$

$$v_o(r, t) = \frac{1}{r}\int_0^r \omega(r', t)r'dr' = \frac{\Gamma}{2r}\left(1 - \exp\left\{-\frac{r^2}{4\Sigma(\nu, t, \sigma)}\right\}\right). \tag{66}$$

Here $\Gamma$ is the integration constant having dimension [length$^2$/time] and the denominator $\Sigma(\nu, t, \sigma)$ has a view

$$\Sigma(\nu, t, \sigma) = \int_0^t \nu(\tau)d\tau + \sigma^2. \tag{67}$$

Here $\sigma$ is an arbitrary constant such that the denominator is always positive.

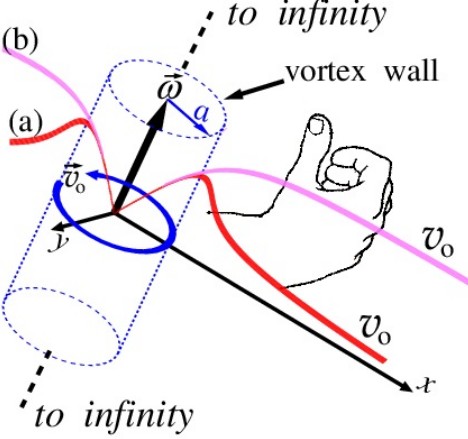

**Figure 2.** Vortex shown in the cylindrical coordinate system extends from minus infinity to plus infinity. The vortex wall is a boundary, where the orbital speed, $v_o$, reaches maximal values; $a$ is the radius of the vortex tube, $\vec{\omega}$ is the vorticity. Profiles of the orbital velocity $\vec{v}_o$ at different $N$ in Equation (25): (**a**) $N = 1$, normal profile; (**b**) $N \gg 1$, nondecreasing profile with distance.

Long-Lived Coherent Gaussian Vortices

One can see from Equation (67) that when $t$ tends to infinity we get $\langle \mu(t) \rangle = 0_+$. In this case solutions (65) and (66) degenerate to the so-called long-lived Gaussian vortices [79–81]. Further, we will deal with the orbital velocity (66). It gets the following view

$$v_o(r, \sigma) = \frac{\Gamma}{2r}\left(1 - \exp\left\{-\frac{r^2}{4\sigma^2}\right\}\right).$$
(68)

The coherent Gaussian vortices being time-independent objects allow superposition:

$$v(r) = \sum_{n=1}^{N} v(r, \sigma_n),$$
(69)

where $\sigma_n$ grows with increasing $n$. For the sake of simplicity, here we consider the following simple dependence

$$\sigma_n = \sigma \cdot n = 0.1n, \quad n = 1, 2, \cdots, N.$$
(70)

Figure 2 shows the orbital speed as a function of the distance $r$ for two different cases: $N = 1$, and $N \gg 1$, respectively. The profile of the speed grows monotonically as $r$ increases from the center of rotation. In the case of $N = 1$, the speed after reaching the vortex wall begins to drop monotonously, Figure 2a. While for the case of large enough $N$, the speed behind the vortex wall comes to a flat level, Figure 2b. This case agrees well with astronomical observations of the orbital speeds of the rotating spiral galaxies [16]. Figure 3 shows the orbital speeds for different $N$ as functions of time.

The Helmholtz's theorems state the follows [82]:

**1.** The strength of a vortex filament is constant along its length.
**2.** A vortex filament cannot end in a fluid medium; it must extend to the boundaries of the fluid or form a closed path.
**3.** In the absence of rotating external forces, an initially irrotational fluid remains irrotational.

According to Helmholtz and Kelvin's theorems, the vortex once formed can exist for a very long time. Its opposite endings (a) either go onto infinity, or (b) they end on some boundaries, or (c) glue with each other that form a torus. Note that the emergence of a vortex is accompanied by the appearance of its twin, with the opposite chirality. Their total chirality has to stay zero.

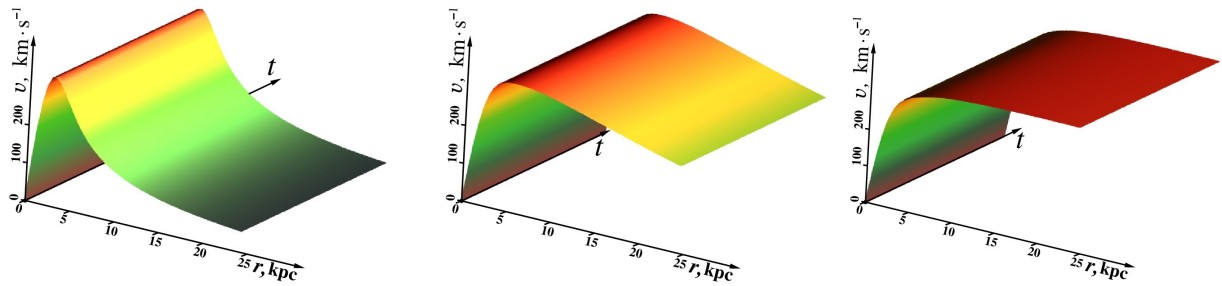

**Figure 3.** Orbital velocity (68) and (69) for $N = 1$, $N = 10$, $N = 100$ from the left to right, respectively.

*4.3. Nonrelativistic Sector: Schrödinger Equation*

Now let us glance on Equation (52) taking into consideration the expressions for the gravitomagnetic field $\vec{\mathcal{E}}$ and the divergence of the force field that are given in (27) and (44), respectively. In this case Equation (52) takes the following view

$$\frac{\partial \vec{p}}{c \partial t} + \nabla \epsilon = \vec{F} + \vec{\Omega}_C$$
(71)

Here $\vec{F}$ is the sum of all force densities acting on the medium under consideration within the unit volume. They are (a) external forces, (b) internal forces represented by the pressure gradients arising in the medium under the action of the external forces, (c) dissipative losses because of the bulk viscosity induced by shear stresses within the medium. The term $\vec{\Omega}_C$ is an arbitrary force density subject to the condition $(\nabla \cdot \vec{\Omega}_C) = 0$. That is, it can be an arbitrary torsion force.

By explicitly writing out terms $\vec{p}$ and $\epsilon$ in nonrelativistic limit given in Equations (17) and (18) we come to the Navier–Stokes equation with some modified terms [78]

$$\rho_M \left( \frac{\partial}{\partial t}\tilde{v} + \overbrace{(\tilde{v} \cdot \nabla)\tilde{v}}^{(a)} \right) + \rho_e \vec{E} = \vec{f}_1 + \vec{f}_2 - \rho_M \nabla (P/\rho_M) + \mu(t)\nabla^2 \tilde{v}, \tag{72}$$

Here $\vec{p}$ and $\epsilon$ in Equation (71) are loaded by the electromagnetic potentials $\vec{A}$ and $\Phi$ as shown in Equation (20). Computation of their contributions in the left part of Equation (71) loads to appearance of the electric field $\vec{E}$ (30) that is written down explicitly in Equation (72).

As for the right part of Equation (72) here the first force term is a conservative $\vec{f}_1(\vec{r}, t) = -\rho \nabla U(\vec{r}, t)$, where $U(\vec{r}, t)$ is the external potential. While the second force term, $\vec{f}_2$, is the Lorentz force [63]

$$\vec{f}_2 = (\rho_e \vec{E} + \vec{J}_e \times \vec{B})/2 = (\rho_e \vec{E} - \rho_e[\vec{B} \times \vec{v}])/2. \tag{73}$$

Here $\rho_e = q/\Delta V = eN/\Delta V = e\rho$ and $q = eN$ is a charge per the unit volume $\Delta V$ ($e$ is the electron charge), $\vec{J}_e = \vec{v}q/\Delta V = \rho_e \vec{v}$ is the density current. $\vec{E}$ is an electric field, and $\vec{B}$ is a magnetic field. Here we take into account $\vec{J}_e \times \vec{B} = -\vec{B} \times \vec{J}_e = -\rho_e \vec{B} \times \vec{v}$. Note that the Lorentz force here enters with a divisor 2. Looking ahead we note that, accurate to a dimensionless factor called the $g$-factor, this divisor gives a value of the gyromagnetic ratio $\gamma = e/2m$ [83].

Overwriting Equation (72) in the light the above said we have

$$m \left( \frac{\partial}{\partial t}\tilde{v} + \overbrace{\frac{1}{2}\nabla v^2 + \underbrace{[\vec{\omega} \times \vec{v}]}_{(b)}}^{(a)} \right) = -\underbrace{\frac{e}{2}[\vec{B} \times \vec{v}]}_{(b)} - \frac{e}{2}\vec{E} - \nabla U - \nabla Q + m\nu(t)\nabla^2 \tilde{v}, \tag{74}$$

Both parts of the equation are divided on the density distribution $\rho = \rho_M/m$, Equation (19). For that reason instead of the dynamical viscosity coefficient $\mu$ we write here the kinematic viscosity coefficient $\nu = \mu/\rho_M$. A contribution of the electric field $(e/2)\vec{E}$ is related to the gradient of the external potential $U$ by redefinition of the latter $U \rightarrow U - (e/2)\Phi$. As for the terms covered by brace (a) they come from $(\vec{v} \cdot \nabla)\vec{v}$ represented in Equation (72). The first term under the brace is the kinetic energy and the vector $\vec{\omega} = [\nabla \times \vec{v}]$ is called vorticity.

A term $Q$ in Equation (74) is the quantum potential that is written instead of the term $P/\rho$ shown in Equation (72) as $P/\rho_M$. Here we need to note that the term $\rho_M \nabla(P/\rho_M)$ can be rewritten as

$$\rho_M \nabla \left( \frac{P}{\rho_M} \right) = \rho \nabla \left( \frac{P}{\rho} \right) = \nabla P - P \nabla \ln(\rho). \tag{75}$$

Only the first term from the right in this expression, the pressure gradient $\nabla P$, is represented in the original classical Navier–Stoke equation [7,82]. The second term, $P\nabla \ln(\rho)$, is an added term describing the change in the logarithm of the density distribution $\rho$ on the infinitesimal increment of length multiplied by $P$. From here it follows that the modified pressure gradient is conditioned by the classical pressure gradient and by adding the entropy gradient, $\ln(\rho)$ [84], multiplied by the pressure. The latter term represents itself as an increment of the information flow per length multiplied by the pressure $P$ that

introduces a pure quantum effect. Let us show this. With this aim, first, we suppose that the pressure $P$ is the sum of two ones, $P_1$ and $P_2$.

As for the pressure $P_1$, we begin with Fick's law, which says that the diffusion flux, $\vec{J}$, is proportional to the negative value of the mass density gradient, $\vec{J} = -D\nabla\rho_M$. Here, $D$ is the diffusion coefficient. Since $D\nabla\vec{J}$ has the dimension of the pressure, we define the first pressure:

$$P_1 = D\nabla\vec{J} = -D^2\nabla^2\rho_M. \tag{76}$$

Observe that the kinetic energy of the diffusion flux of the fluid medium is $(M/2)(\vec{J}/\rho_M)^2$. This means that one more pressure exists as the average momentum transfer per unit volume:

$$P_2 = \frac{\rho_M}{2}\left(\frac{\vec{J}}{\rho_M}\right)^2 = \frac{D^2}{2}\frac{(\nabla\rho_M)^2}{\rho_M}. \tag{77}$$

One can see that the sum of the two pressures, $P_1 + P_2$, divided by $\rho$ (we remark that $\rho_M = m\rho$, see Equation (19)) reduces to the quantum potential [74]

$$Q = \frac{P_2 + P_1}{\rho} = m\frac{D^2}{2}\left(\frac{\nabla\rho}{\rho}\right)^2 - mD^2\frac{\nabla^2\rho}{\rho} = -2mD^2\frac{\nabla^2 R}{R}. \tag{78}$$

Here $R = \sqrt{\rho}$ is the amplitude distribution of the carriers of $m$ in the volume $\Delta V$.

Now we need to define the diffusion coefficient $D$ in terms related to the quantum mechanics problems. These problems were considered by E. Nelson in his monographs [9,10]. As follows from his article [8] Brownian motion of a carrier of the single mass in the quantum ether is described by the Wiener process with the diffusion coefficient equal to

$$D = \frac{\hbar}{2m}, \tag{79}$$

see Equation (4). As a result, the quantum potential in the nonrelativistic limit is as follows:

$$Q = \frac{\hbar^2}{2m}\left(\frac{\nabla\rho}{\rho}\right)^2 - \frac{\hbar^2}{4m}\frac{\nabla^2\rho}{\rho} \tag{80}$$

Let us now bring the Equation (74) to the modified Hamilton–Jacobi equation by expressing first the velocity $\vec{v}$ in terms of the gradient of the action function $S$.

$$\vec{p} = m\vec{v} = \nabla S + m\vec{v}_o, \qquad m\frac{v^2}{2} = \frac{1}{2m}(\nabla S)^2 + m\frac{v_o^2}{2} \tag{81}$$

Here $\vec{v}_o$ is the orbital velocity. Now, as for the terms covered by braces (b) in Equation (74), we believe that they cancel each other. In other words, we accept the following agreement

$$\vec{\omega} + \frac{e}{2m}\vec{B} = 0. \tag{82}$$

After remarks made above we rewrite Equation (74) as follows

$$\frac{\partial}{\partial t}(\nabla S + \underbrace{m\vec{v}_o}_{(c)}) + \frac{1}{2m}\nabla\big((\nabla S)^2 + \underbrace{m^2 v_o^2}_{c}\big) + \nabla(Q + U) - \underbrace{m\nu(t)\nabla^2\vec{v}}_{d} = 0. \tag{83}$$

First of all, one can note the two terms embraced by braces (c) relate to the conservation law of vortex filaments formulated by the first Helmholtz's theorem (placed above Figure 3). It reads

$$\frac{\partial}{\partial t}m\vec{v}_o + \frac{1}{2m}\nabla m^2 v_o^2 = 0 \;\Rightarrow\; \frac{\partial}{\partial t}\vec{v}_o + [\vec{\omega}\times\vec{v}_o] = 0. \tag{84}$$

Here $\vec{\omega} = [\nabla \times \vec{v}_o]$ is the vorticity. The equation from the right is the continuity equation for the velocities field on the sphere $\mathbb{S}^3$ of unit radius. In particular, replacing $\vec{\omega}$ by the magnetic induction $\vec{B}$ multiplied by the gyromagnetic ratio $\gamma = (e/2m)$, as shown in Equation (82), and expressing the velocity $\vec{v}_o$ through the 4D spinor representation we come to the equation describing the behavior of a spin in the magnetic field [59].

Notice that the term embraced by a brace (d) reads as $\vec{v} = \vec{v}_S + \vec{v}_o = (1/m)\nabla S + \vec{v}_o$. Further, there is a one-to-one correspondence between the hydrodynamical and electromagnetic variables [85]. This correspondence permits the expansion of $\nabla S$ in Equation (83) to $\nabla S - e\vec{A}$, where $\vec{A}$ is the vector potential. As a result, we come to the following modified Hamilton–Jacobi equation [63]

$$\frac{\partial}{\partial t}S + \frac{1}{2m}(\nabla S - e\vec{A})^2 + \frac{m}{2}v_o^2 + (Q + U) - \underbrace{\nu(t)\nabla^2 S}_{(d)} = C_1. \tag{85}$$

Here $C_1$ is the integration constant. For completeness, let us add two continuity equations

$$\frac{\partial}{\partial t}\rho_M + (\nabla \cdot \tilde{v})\rho_M = 0, \tag{86}$$

$$\frac{d}{dt}|\varphi(t)\rangle = -\frac{\mu_B}{\hbar}(\vec{\eta} \cdot \vec{B})|\varphi(t)\rangle. \tag{87}$$

The second equation describes the continuity of the ket-vector flow, $|\psi\rangle$, on the surface of the unit sphere $\mathbb{S}^3$. Here $(\vec{\eta} \cdot \vec{B}) = -\mathrm{Re}\,\mathbb{F}_{\mathcal{E}\mathcal{B}}$ is $4 \times 4$ matrix of the magnetic induction $\vec{B}$ as shown in Equation (25), and $\mu_B = e\hbar/2m$ is the Bohr magneton. The ket vector $|\varphi(t)\rangle$ contains four real variables $s_0, s_x, s_y, s_z$ satisfying the condition $s_0^2 + s_x^2 + s_y^2 + s_z^2 = 1$ [63].

One can see that the term enclosed by brace (d) in Equation (85) is rewritten as follows $\nabla^2 S = m\nabla(1/m)\nabla S = m(\nabla \cdot \vec{v})$. Further we can replace the term $(\nabla \cdot \vec{v})$ by $-(\partial/\partial t)\ln(\rho_M)$ what follows from (86). Finally, we note that the Hamilton–Jacobi Equation (85) together with the continuity Equations (86) and (87) can be extracted from the following Schrödinger-Pauli-like equation [63]

$$\mathbf{i}\hbar\frac{\partial}{\partial t}|\Psi(\vec{r},t)\rangle = \underbrace{\left( \frac{1}{2m}\left( -\mathbf{i}\hbar\nabla + m\vec{v}_o - e\vec{A} \right)^2 + U(\vec{r}) + \overbrace{\nu(t)m\frac{\partial}{\partial t}\ln\left(|\Psi|^2\right)}^{(d)} \right)|\Psi(\vec{r},t)\rangle}_{\text{Schrodinger}-\text{like equation}}$$

$$- \underbrace{\mathbf{i}\mu_B(\vec{\eta} \cdot \vec{B}(\vec{r},t))|\Psi(\vec{r},t)\rangle}_{\text{Stern}-\text{Gerlach term}} + C_1|\Psi(\vec{r},t)\rangle \tag{88}$$

as soon as we represent the wave function beforehand in the polar form

$$|\Psi(\vec{r},t)\rangle = \sqrt{\rho(\vec{r},t)}|\varphi(t)\rangle \cdot \exp\{\mathbf{i}S(\vec{r},t)/\hbar\}. \tag{89}$$

and substitute this function into the Schrödinger-Pauli equation. After separating variables into imaginary and real components we come to Equations (85)–(87).

In particular, one can conclude from here that the complex wave function describes a physical process occurring in the quantum ether. The term $\rho(\vec{r},t)$ gives the probability density of detecting a particle in the vicinity of point $\vec{r}$ at the moment $t$. While the action function $S(\vec{r},t)$ shows the mobility of particle in the vicinity of the same point $\vec{r}$ at the moment $t$. Note that according to the uncertainty principle (2), the mobility of the particle and its position are two mutually exclusive quantities that cannot be measured simultaneously. Note also that the term covered by a brace (d) in Equation (88) represents itself as a source of color noise because of its nonlinearity. For that reason, this equation can be named the Schrödinger-Pauli-Langevin equation.

Ginsburg–Landau Theory of Super-Fluidity

The theory of dark matter superfluidity represented by Bose–Einstein condensate is discussed widely in the open press [1,24,26,28,35,76,86]. Here we consider this issue in the plan of the Ginzburg–Landau theory of superconductivity [87]. Let the superconducting carriers $n_s := \rho_s = |\psi|^2$ be the order parameter. Let us for that first write the stationary Equation (88):

$$\underbrace{\frac{1}{2m}\left(-\mathbf{i}\hbar\nabla + m\vec{v}_{\mathrm{o}} - e\vec{A}\right)^2|\psi\rangle - \mathbf{i}\mu_B(\overrightarrow{\eta}\cdot\vec{B}(\vec{r},t))|\psi\rangle}_{(c)}$$

$$+ \; \underbrace{\nu(t)m\frac{d}{dt}\ln(|\psi|^2)|\psi\rangle}_{(a)} \; + \; \underbrace{a|\psi\rangle + b|\psi|^2|\psi\rangle}_{(b)} \; = \; E|\psi\rangle. \tag{90}$$

The term, $a + bn_s$, enclosed in brace (b) stems from decomposition of the potential $U(\rho(\vec{r}))$ into the Taylor series. In this case, $n_s$ characterizes the number of bosons populating the superconducting fraction. Ibid $E|\psi\rangle = \mathbf{i}\hbar\partial|\psi\rangle/\partial t$. Note that the potential $U(\rho(\vec{r}))$ we consider in a generalized Gross–Pitaevskii representation [24,25,29,88–90].

Let us multiply Equation (90) by $\langle\psi|$ from the left. We note first that the term enclosed by brace (a) vanishes due to the condition $\langle\nu(t)\rangle = 0_+$ shown in Equation (63). Further we declare that the terms covered by braces (b) and (c) lead to two independent equations:

$$\langle E_{\mathrm{s,pot}}\rangle = a|\psi|^2 + b|\psi|^4, \tag{91}$$

$$\vec{j}_s = \mathbf{i}\frac{\hbar}{2}(\psi^*\nabla\psi - \psi\nabla\psi^*) - m\vec{v}_{\mathrm{o}}\cdot|\psi|^2 + \underbrace{(e\vec{A} - \overbrace{\vec{\mu}_s\cdot(\overrightarrow{\eta}\cdot\vec{B}(\vec{r},t))}^{(d)})/c)\cdot|\psi|^2}_{(e)}, \tag{92}$$

and $\langle E_{\mathrm{s,kin}}\rangle = (1/2m)\cdot|j_s|^2$. Brace (d) covers the product of magnetic moment $\vec{\mu}_s$ and magnetic induction $\vec{B}$ written in the quaternion representation (25). Opening this product, we find two forms. One form is a scalar product, $(\vec{\mu}_s\cdot\vec{B})$, that vanishes. The second form is a vector product, $[\vec{\mu}_s\times\vec{B}]$. By definition, it is the torque $\vec{\tau}$. There is a phenomenon known as the Meissner Effect [31]. There is the complete displacement of the magnetic field from the conductor volume during its transition to a superconducting state. For that reason, we will assume the $e\vec{A} = \vec{\tau}/c$ in the superfluid.

The Equations (91) and (92) are like those of the Ginzburg–Landau theory [91]. Here the contribution of magnetic induction, $\vec{B}$, to the kinetic energy $\langle E_{\mathrm{s,kin}}\rangle$ reads $\mu_B|B|$.

Figure 4 shows solutions of Equation (91) with a positive parameter $b$. While the parameter $a$ can have either positive value, Figure 4a, or negative one, Figure 4b. In the first case there is no superfluid component, while in the second case it appears. Its density distribution is $\rho_{\mathrm{s}} = |\psi|^2 = -a/(2b)$.[1] The value $a = 0$ represents the bifurcation point dividing the normal phase ($a > 0$) from the superfluid one ($a < 0$). This parameter has the following representation

$$a = k_{\mathrm{B}}(T - T_\lambda). \tag{93}$$

Here $k_{\mathrm{B}}$ is the Boltzmann parameter and $T_\lambda$ is the critical $\lambda$-point. Note that the parameters $a$ and $b$ have dimension [energy].

---

[1] for simplicity, the density distribution is written in dimensionless view.

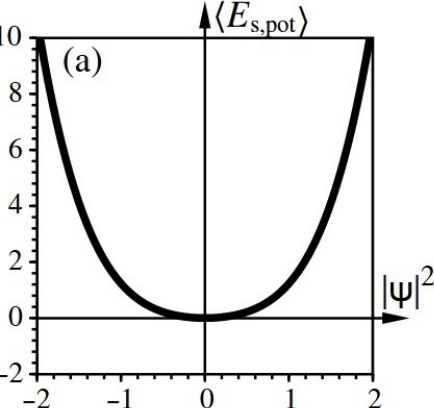 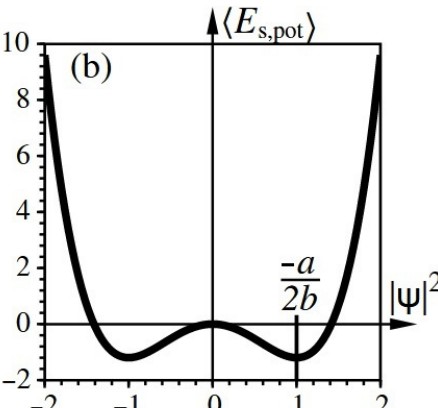

**Figure 4.** Function $\langle E \rangle$ versus $|\psi|^2 = \rho_s$: (**a**) the parameter $a > 0$; (**b**) the parameter $a < 0$. The point $-a/(2b)$ is position of energy minimum of the function $\langle E_{s,pot} \rangle$ written out in Equation (91).

Below the lambda line (below $T_\lambda$, Figure 5), we can phenomenologically describe the superfluid by the so-called two-fluid model. It manifests itself as if it consists of two components: a normal one, which behaves like an ordinal liquid, and a superfluid component with zero viscosity and zero entropy [92]. The ratio of the density of normal ($\rho_n$) and superfluid ($\rho_s$) components depends on the temperature shown in Figure 5. As the temperature decreases, the fraction of density of the superfluid component ($\rho_s/\rho$) increases from zero at $T_\lambda$ to one at zero Kelvin temperature.

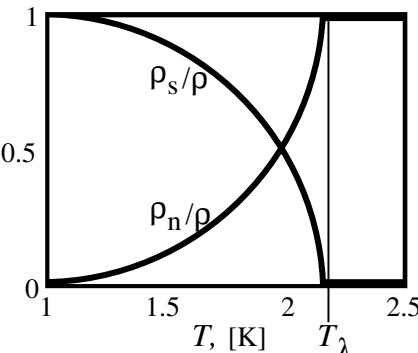

**Figure 5.** Temperature dependence of the relative superfluid and normal components, $\rho_s/\rho$ and $\rho_n/\rho$, as functions of $T$. $T_\lambda$ is the critical $\lambda$-point [93].

Note that the dispersion of the viscosity coefficient, $\langle \nu(t)\nu(0) \rangle > 0$, says that there is a permanent exchange of information between the normal and superfluid components. We note first if the information transfer within the normal component is not more than the diffusion length that in the superfluid component, it tends to infinity due to the absent dissipation. This critical remark relates directly to the quantum potential. From here it follows that the quantum potential can transmit information for no more than the diffusion length $\sqrt{(\hbar/2m) \cdot \tau}$, where $\tau$ is the characteristic transmission time. However, due to the nonzero dispersion of the viscosity coefficient, $\langle \nu(t)\nu(0) \rangle > 0$, there arises the possibility to transmit information on an arbitrarily long distance due to the superfluid component.

Now let us glance to Equation (92) with discarded terms under the brace (e). This equation answers the question: what should the value of the flux quantum be? Here we repeat calculations made by Dr. Vasiliev in [91]. First, we write out the wave function in the polar form:

$$|\psi(\vec{r})\rangle = \sqrt{n_s}e^{\,\mathbf{i}(S(\vec{r})/\hbar + \theta(\vec{r})/2)}, \tag{94}$$

where $n_s$ is the density of super-fluidity carriers, $S$ is the action function, and dimensionless $\theta$ is the order parameter phase. Here this parameter has divisor 2 specifying that the carrier

possesses the half-integer spin. Note that $\exp\{i\theta/2\}$ is a factor of the Pauli matrices belonging to SU(2).

If the orbital velocity, $\vec{v}_o$, is zero, then we omit $\theta$. In this case, the density of particle flux we describe by the equation

$$mn_s\vec{v}_s = i\frac{\hbar}{2}(\psi\nabla\psi^* - \psi^*\nabla\psi) = n_s\nabla S = n_s m\vec{v}_s. \tag{95}$$

Here the leftmost $\vec{v}_s$ is the velocity of superfluid carriers, while in the rightmost $\vec{v}_s = (1/m)\nabla S$ the subscript s means an irrotational velocity [42]. From here it follows, that the superfluid carriers have only irrotational velocities.

Now let us return to Equation (92) and substitute to this equation the wave function (94). We get

$$\underbrace{mn_s\vec{v}_s = n_s m\vec{v}_s}_{(a)} + \underbrace{\frac{\hbar}{2}\nabla\theta - mn_s\vec{v}_o}_{(b)}. \tag{96}$$

The terms covered by a brace (a) annihilate each other. As for the expression covered by a brace (b), we rewrite out it by the following equation

$$\nabla\theta = \frac{2m}{\hbar}n_s\vec{v}_o. \tag{97}$$

Since $\vec{v}_o$ is the orbital velocity, we take the integral over a closed contour about the axis perpendicular to the plane of rotation:

$$\underbrace{\oint \nabla\theta ds}_{(c)} = \frac{2m}{\hbar}n_s\oint \vec{v}_o ds = \frac{2m}{\hbar}\Phi, \tag{98}$$

where

$$\Phi = n_s\oint \vec{v}_o ds \tag{99}$$

is the vorticity flux through any loop.

The contour integral covered by brace (c) in (98), must be a multiple of $2\pi$, to ensure the uniqueness of the order parameter in a circuit along the path. Thus, the vorticity flux trapped by the super-fluidity ring should be multiple to the quantum of vorticity flux:

$$\Phi_m = \frac{h}{2m}. \tag{100}$$

Put attention that here $h = 2\pi\hbar$ is the Planck constant, while in Equation (98) we write $\hbar$, the reduced Planck constant. We note that accurate to the divider 2, this quantity is shown in Lounasmaa and Thuneberg article [94] as $\kappa_0 = h/m$ and is named the circulation quantum.

The dimensionality of the vorticity flux quantum is m$^2$/s. It is the same as for the diffusion coefficient and the kinematic viscosity coefficient. It is not a random coincidence. All these constants describe the dissipation of energy or heat through the area unit per the unit of time. Superfluid materials displace away or encapsulate similar inclusions by forming separate vortex filaments. It is known as the Meissner effect. This effect explains the expulsing magnetic field from the superconductor at its transition to the superconducting state. It makes the superfluid material invisible to many forms of interaction.

Observe that the vorticity flux quantum (100) in the denominator contains the doubled mass. The superfluidity is because its carriers have mass $2m$, i.e., they represent two paired particles. Note that the wave function contains the added phase $\theta$ divided by 2, which points to the carriers with half-integer spins. It leads to the carrier pairing. In the original

phenomenological theory, this divisor absents [91]. However, instead, the multiplier 2 is present in the term $2(e/c)\vec{A}$.

Observe that the temperature of CMB is about 2.725 K [11], which is also the temperature of the thermal black body radiation. Since the predominant atom in outer space is hydrogen, we take proton as a unit of the superfluid quantum ether. Two carriers, proton plus antiproton, can create a boson with a zero charge and an integer spin. The mass of the boson is twice the proton mass.

If the paired particles are two protons, then two electrons should accompany to neutralize charges. If they are proton and antiproton, then their interaction should occur through the zero-point vacuum fluctuations [91]. Between these particles, electron-positron virtual pairs arise and annihilate, providing the self-sustained buffer lattice [95]. The Feynman path integral technique proposes a diagram shown in Figure 6. The electron-positron pairs provide the superfluidity effect that is like Cooper pairs due to a correlated state of electrons with opposite spins and momenta.

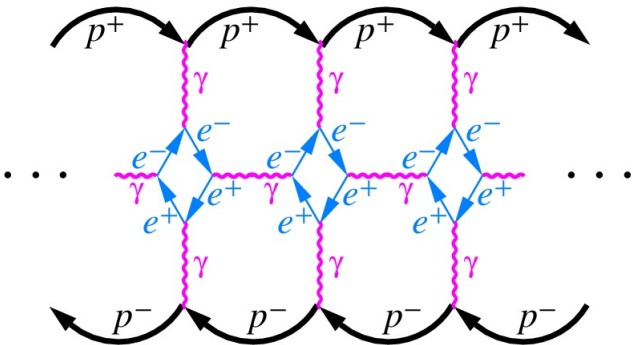

**Figure 6.** Feynman's diagram of boson $(p^+p^-)$ consisting of proton $p^+$ and antiproton $p^-$ coupled by exchanging $\gamma$-quanta through electron-positron interactions. The latter provides the superfluidity effect that is like the Cooper pairs due to a correlated state of electrons with opposite spins and momenta.

In the case of the particle and antiparticle with opposite charges, the pairing by the zero-point fluctuations should provide their repulsion in order to prevent the annihilation. Like an electron being on a stationary orbit does not drop on a proton in the hydrogen atom, the particle-antiparticle complex can form a long-lived orbit. The proton, $p^+$, and antiproton, $p^-$, can unite into triplet orto-boson with three spin projections, $s = 1, 0, -1$ of the total integer spin $I = 1$. The corresponding wavefunctions are $|\uparrow\uparrow\rangle$, $(1/2)(|\uparrow\downarrow\rangle + |\downarrow\uparrow\rangle)$, $|\downarrow\downarrow\rangle$. Or they can unite into singlet paraboson with the total integer spin $I = 0$ and the wavefunction $(1/2)(|\uparrow\downarrow\rangle - |\downarrow\uparrow\rangle)$.

The principal condition for the superfluid state formation is the ordering of zero-point oscillations [91]. Due to this effect, the proton-antiproton pairs attract each other by forming the Bose–Einstein condensate. In application to the particle-antiparticle's pair, let us adopt the Bohr radius, $a_0$, as a physical constant equal to the most probable distance between the proton and antiproton rotating about the center of mass in their ground state:

$$a_0 = r_1 = \frac{\hbar}{\alpha m c} \approx 1.44 \cdot 10^{-14} \text{ m,} \tag{101}$$

where $\alpha \approx 1/137$ is the fine structure constant. The Bohr radius that is also the radius of the first orbit, $n = 1$, is about sixteen times larger than the radius of the proton. The latter is about $8.8 \cdot 10^{-16}$ m [96].

Following Niels Bohr's assumption, concerning the energy levels and spectral frequencies of the hydrogen atom, we can guess that a proton-antiprotons pair does not annihilate with emitting $\gamma$-quantum, while it is in one of the stationary states. The particles can gain or lose energy by jumping from one discrete orbit to another. The particle-antiparticle pair in the ground state looks like a torus. The torus radius is equal to the Bohr radius, while the

tube radius corresponds to the proton radius. As was stated above, proton and antiproton can form either as orthoboson or as paraboson. They rotate about the main axis of the torus with speed:

$$v_1 = \frac{\hbar}{m r_1} = \alpha c \approx 2.188 \cdot 10^6 \text{ m} \cdot \text{s}^{-1}. \tag{102}$$

Regardless of masses of particles, the Bohr velocity is a universal constant that is 1000 times less than the light speed. As seen, it is a nonrelativistic speed. On the other hand, the Bohr radius depends on the mass of the particles. The larger the mass, the smaller the radius.

All paths around the center of mass in the ground state are equal to the orbit perimeter $2\pi r_1$. Observe that by multiplying the perimeter by the particle speed, we get the doubled quantum of the vorticity flux

$$2\pi r_1 v_1 = 2\Phi_m. \tag{103}$$

In particular, for the case of a stationary orbit with the number $n$, we have

$$2\pi r_n v_n = 2n\Phi_m. \tag{104}$$

This formula says that each nonradiating Bohr orbit contains an integer number of pairs of the vorticity flux quanta. From here it follows that there may put off the proton and the antiproton annihilation. The ground state containing only two vorticity flux quanta represents the zero-point energy level of the superfluid medium. On the other hand, the $n$th Bohr orbits are excited states of the proton-antiproton pairs (bosons) induced, for example, by powerful electric discharges. Such a macroscopic clot of the excited bosons can represent by itself the ball lightning. Indirect confirmation is the results set out in the article [97].

*4.4. Relativistic Sector: The Dirac Equation*

Dr. Halil Güveniş in his article [98] demonstrated a perfect derivation of the quantum hydrodynamic equations from the original Dirac equation. By invoking the wave function written in the polar form, he gets two equations for two real functions. The first equation describes the mass density conservation. Furthermore, the second equation, loaded by the relativistic quantum potential, gives the velocities field. Note this pair of equations returns two fermion fluids. The first equation describes the real fermion fluid. In addition, the second deals with the fermion antifluid. Both liquids complementing each other give a single Bose–Einstein condensate.

First, we note that the Dirac equation

$$\gamma^\mu (i\hbar \partial_\mu - e A_\mu)\Psi - mc\Psi = 0 \tag{105}$$

describes the behavior of fermions with spin-1/2 in the external electromagnetic field. Here we use SI units for the potential $A_\mu = (\Phi/c, A_x, A_y, A_z)$ and $\partial_\mu = (c^{-1}\partial_t, \partial_x, \partial_y, \partial_z)$. In this equation, the gamma matrices are written in terms of the $2 \times 2$ identity matrix $\sigma_0$ and of three $2 \times 2$ matrices composed of the Pauli matrices $\sigma_x, \sigma_y, \sigma_z$ (10):

$$\gamma^0 = \begin{pmatrix} \sigma_0 & 0 \\ 0 & -\sigma_0 \end{pmatrix} \quad \gamma^1 = \begin{pmatrix} 0 & \sigma_x \\ -\sigma_x & 0 \end{pmatrix} \quad \gamma^2 = \begin{pmatrix} 0 & \sigma_y \\ -\sigma_y & 0 \end{pmatrix} \quad \gamma^3 = \begin{pmatrix} 0 & \sigma_z \\ -\sigma_z & 0 \end{pmatrix}. \tag{106}$$

The fundamental equation of relativistic quantum mechanics at the existence of the external electromagnetic fields $\Phi$ and $\vec{A}$ in article [98] looks as follows

$$i\hbar \frac{\partial}{\partial t} \begin{pmatrix} \psi \\ \chi \end{pmatrix} - e\Phi \begin{pmatrix} \psi \\ \chi \end{pmatrix} = c(\vec{\sigma} \cdot \vec{\pi}) \begin{pmatrix} \chi \\ \psi \end{pmatrix} + mc^2 \begin{pmatrix} \psi \\ -\chi \end{pmatrix} \tag{107}$$

Here $\psi$ and $\chi$ are two-component spinors composing four-component $\Psi$. Further $\vec{\pi} = -i\hbar\nabla - (e/c)\vec{A}$, and $\vec{\sigma} = (\sigma_x, \sigma_y, \sigma_z)$ are well-known $2 \times 2$ Pauli matrices.

- To perform a decoupling of the spinors $\psi$ and $\chi$ we first resolve the spinor $\chi$ with respect $\psi$ *in the upper row* of Equation (107):

$$c(\vec{\sigma} \cdot \vec{\pi})\chi = i\hbar \frac{\partial}{\partial t}\psi - e\Phi\psi - mc^2\psi. \tag{108}$$

Now, by multiplying the *lower row* of Equation (107) by $c(\vec{\sigma} \cdot \vec{\pi})$ we obtain

$$i\hbar \frac{\partial}{\partial t}c(\vec{\sigma} \cdot \vec{\pi})\chi = c^2(\vec{\sigma} \cdot \vec{\pi})(\vec{\sigma} \cdot \vec{\pi})\psi + e\Phi c(\vec{\sigma} \cdot \vec{\pi})\chi - mc^2 c(\vec{\sigma} \cdot \vec{\pi})\chi \tag{109}$$

Further, in this equation we replace all $c(\vec{\sigma} \cdot \vec{\pi})\chi$ by the result obtained in Equation (108). As a result we come to a separate equation for the spinor $\psi$. By repeating the computations of Equations (108) and (109) with respect the spinor $\psi$ versus $\chi$ we also get a separate equation for the spinor $\chi$.

As a result of the above computations, we get two uncoupled equations for both spinors:

$$\left(i\hbar\frac{\partial}{\partial t} - e\Phi\right)^2 \psi = c^2\left(-i\hbar\nabla - \frac{e}{c}\vec{A}\right)^2 \psi - e\hbar c(\vec{\sigma} \cdot \vec{B})\psi + (mc^2)^2\psi, \tag{110}$$

$$\left(i\hbar\frac{\partial}{\partial t} - e\Phi\right)^2 \chi = c^2\left(-i\hbar\nabla - \frac{e}{c}\vec{A}\right)^2 \chi - e\hbar c(\vec{\sigma} \cdot \vec{B})\chi + (mc^2)^2\chi. \tag{111}$$

Here we have replaced $(\vec{\sigma} \cdot \vec{\pi})(\vec{\sigma} \cdot \vec{\pi}) = \pi^2 - e(\hbar/c)(\vec{\sigma} \cdot \vec{B})$, $\vec{B} = [\nabla \times \vec{A}]$, and $(\nabla \cdot \vec{A}) = 0$. Note that Equations (110) and (111) for both spinors, $\psi$ and $\chi$, are absolutely equivalent. This is an amazing fact that hints that two equivalent fermions can be paired in such a way that they form a long-lived Bose particle, Figure 7. It is said in this case that a particle that is its own antiparticle is called a Majorana fermion [99].

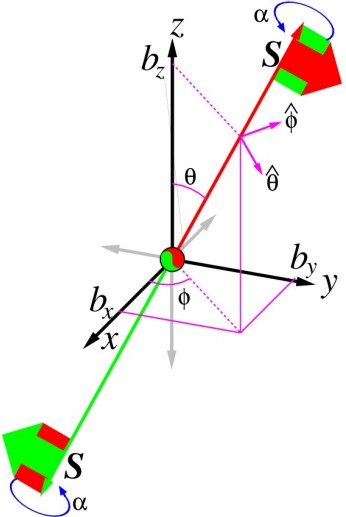

**Figure 7.** The CPT invariance is the fundamental symmetry of physical laws under transformations involving simultaneous inversion of charge, parity, and time. The CPT theorem proves a strict correspondence between matter and antimatter. In particular, a particle and an antiparticle have the same mass and magnetic moment. Their electric charges are equal in modulus and opposite in sign, and their spins are equal in modulus and opposite in direction. According to the CPT theorem, two paired fermions with mirror half-integer spins $S$ can form a long-lived Bose-particle with integer spin. Here the half-integer spin $s = \cos(\alpha/2) + (b_x e_x + b_y e_y + b_z e_z)\sin(\alpha/2)$ can be seen as a pseudovector $(b_x, b_y, b_z)$ equipped with a flag that turns around the flagstaff on an angle $\alpha$.

To come from Equations (110) and (111) to the hydrodynamic representations of these Majorana fermion fields, first we set the spinors $\psi$ and $\chi$ in the polar form

$$\begin{pmatrix} \psi \\ \chi \end{pmatrix} = \begin{pmatrix} \sqrt{\rho}\exp\{iS/\hbar\} \\ \sqrt{\rho}\exp\{iS/\hbar\} \end{pmatrix}. \tag{112}$$

Since two fermion fields are mirror-symmetric and equivalent to each other, we do not introduce differences through the density distribution $\rho$ and the action function $S$.

By repeating the path paved by Madelulg [100], de Broglie [38], Bohm [101,102], we finally obtain two equations for the imaginary and real parts

$$-\frac{\partial}{\partial t}\left( \frac{\rho}{c^2}\left( \frac{\partial S}{\partial t} + e\Phi \right) \right) + \nabla(\rho \cdot (\nabla S - (e/c)\vec{A})) = 0, \tag{113}$$

$$\frac{1}{c^2}\left( \frac{\partial S}{\partial t} + e\Phi \right)^2 = (mc)^2 + (\nabla S - (e/c)\vec{A})^2 - e(\hbar/c)(\vec{\sigma}\cdot\vec{B}) - \hbar^2\frac{\Box\sqrt{\rho}}{\sqrt{\rho}}, \tag{114}$$

By substituting the new variables

$$\vec{p} = \nabla S \qquad \text{and} \qquad E = -\frac{\partial S}{\partial t} \tag{115}$$

into Equations (113) and (114) we obtain

$$\frac{\partial}{\partial t}\left( \frac{\rho}{c^2}(E - e\Phi) \right) + \nabla(\rho \cdot (\vec{p} - (e/c)\vec{A})) = 0, \tag{116}$$

$$(E - e\Phi)^2 = (\vec{p}c - e\vec{A})^2 + m^2c^4 - e\hbar c(\vec{\sigma}\cdot\vec{B}) - \hbar^2c^2\frac{\Box\sqrt{\rho}}{\sqrt{\rho}}, \tag{117}$$

$$\oint \left( \vec{p} - \frac{e}{c}\vec{A} \right) d\vec{r} = 2\pi\hbar m, \qquad m = 0, \pm 1, \pm 2, \cdots \tag{118}$$

Here Equation (116) is the equation of continuity of the quantum electrohydrodynamical medium, Equation (117) is the equation of motion, and Equation (118) is a quantization rule for the angular momentum going back to Bohr-Sommerfeld-Wilson. The latter equation is added by Halil Güveniş in his following articles [103,104]. These articles aim to make computations, in the first approximation, of possible models of a homogeneously charged proton-antiproton pair.

### 4.4.1. Majorana Fermions: Proton-Antiproton Dancing Ensemble

Dr. Halil Güveniş in his works [103,104] set the task to solve the fundamental equations of quantum electrohydrodynamics in the first approximation of a uniformly charged proton-antiproton pair. The main aim is to compute the characteristic parameters of these particles. They are the energy-mass, radius, and internal pressure of a quantum hydrodynamic medium.

We will not lay out all the calculations here. However, it makes sense to show some of the key points. First we note that the electromagnetic potentials $\Phi$ and $\vec{A}$ occurring in the fundamental Equations (116)–(118) are defined by the equations:

$$\Phi(\vec{r}, t) = -4\pi Ze \int d^3r' \int dt' \rho(\vec{r}', t')G(\vec{r} - \vec{r}', t - t'), \tag{119}$$

$$\vec{A}(\vec{r}, t) = -4\pi Ze \int d^3r' \int dt' \frac{\vec{p}}{m_0c}\rho(\vec{r}', t')G(\vec{r} - \vec{r}', t - t'), \tag{120}$$

where

$$G(\vec{r} - \vec{r}', t - t') = -\frac{\delta(t - t' - |\vec{r} - \vec{r}'|/c)}{4\pi|\vec{r} - \vec{r}'|} \tag{121}$$

is the Green's function and $Ze$ the total charge of the considered quantum hydrodynamic system.

Güveniş solves the equation system (116)–(121) for a proton-antiproton pair and thereby determines the mass or charge density $\rho$. We are interested only in stationary solutions. It means that the mass density $\rho$ must satisfy the boundary condition $\rho(r_0) = 0$ and the equation of continuity (116) reduces to $(\nabla \cdot \rho \vec{p}) = 0$. As a result we get a snapshot of the instant proton-antiproton state. Since the equations for both proton and antiproton, Equations (110) and (111), are equivalent we can conclude that the got snapshot points to the alternating proton-antiproton dancing.

Further, the task we solve in the spherical coordinate system by assuming the amplitude distribution as a function of the radial distance $r$, azimuthal $\theta$, and polar $\phi$ angles:

$$\sqrt{\rho} = f(r)g(\phi, \theta). \tag{122}$$

The solutions will contain the spherical harmonics

$$g(\phi, \theta) = (-1)^m \sqrt{\frac{(l - |m|)!(2l + 1)}{4\pi(l + |m|)!}} e^{im\phi} \frac{1}{2^l l!} \frac{d}{d(\cos(\theta))^l} (\cos^2(\theta) - 1)^l, \tag{123}$$

where $l = 0, \pm 1, \pm 2, \cdots$ and $-l \leq m \leq +l$. As for the function $f(r)$ in Equation (122) it determines a profile of the function $\rho(r)$ along the radial distance $r$ from the origin. After series of computations [103] the density distribution function $\rho(r)$ for the case of antiproton in the ground state looks as

$$\rho(r) = \frac{1}{0.363393} \left( \sin(38.7656 r^3))e^{-0.625r}r^{-1.375} \right)^2 \tag{124}$$

Number coefficients result from calculations given in [103]. Figure 8 shows a view of this function.

Experiments at scattering electrons with a wavelength of $\lambda = 10^{-15}$ m on protons show that the proton is not a point particle but an extended object in the space with an averaged radius about 0.84–0.87 fm. It has no clearly defined boundary surface. It is a loose object where the periodicity of the density along the radius $r$ takes place, as shown in Figure 8.

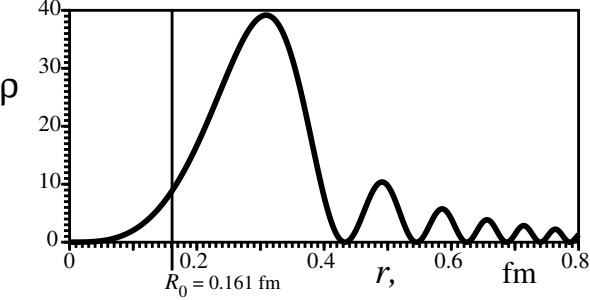

**Figure 8.** Density distribution function $\rho(r)$ versus $r$.

For a detailed consideration of these results, it is appropriate to calculate also the pressure $P_1$ acting on the sphere surface $4\pi r^2$ of the antiproton.

$$
\begin{aligned}
P_1 &= \frac{1}{4\pi} \left[ e^2 a_0 \frac{r^3 - R_0^3}{r} - \sqrt{b^2 + \frac{0.75}{r^2} - \frac{e^2}{2b} a_0 \left(1 + \frac{2R_0^3}{r^3}\right)} \right] \left( \sin\left(e^2 a_0 \frac{r^3}{3}\right) r^{-n} e^{-ar} \right)^2 \\
&= \frac{9.25462 e^{-1.25r}}{94.1501} \left( r^3 - 0.00416282 - r\sqrt{\frac{0.75r - 0.123251}{r^3} + 0.624746} \right) \frac{\sin^2(38.765 r^3)}{r^{3.75}},
\end{aligned} \tag{125}
$$

The parameters $R_0$, $a_0$, $a$, $b$, $n$ and their numerical values are given in [103], $e$ is the electron charge. The pressure $P_1$ as a function of the distance $r$ is shown in Figure 9. The presence in Equation (125) of root with the negative term can be the reason for the emergence of the imaginary pressure. Such an imaginary pressure appears in region of $r < R_0 = 0.161$ fm. The appearance of imaginary solutions is not a defect of the theory, but it can point to a new physical phenomenon. For that reason, let us first look on the pressure $P_1$ in the range $(0, 0.16)$ fm in zoom, Figure 10. One can see that in this region, there is a small real pressure with maximal deviation near $\delta r$ equal to about 0.12 fm.

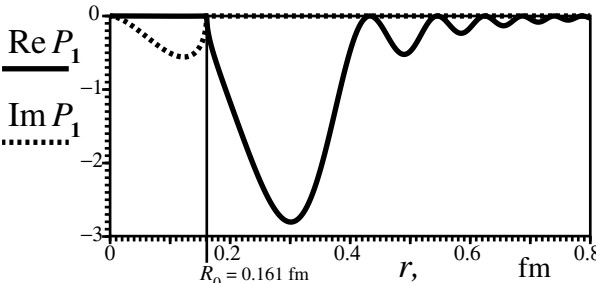

**Figure 9.** Real (solid) and imaginary (dotted) parts of the pressure $P_1(r)$ versus $r$.

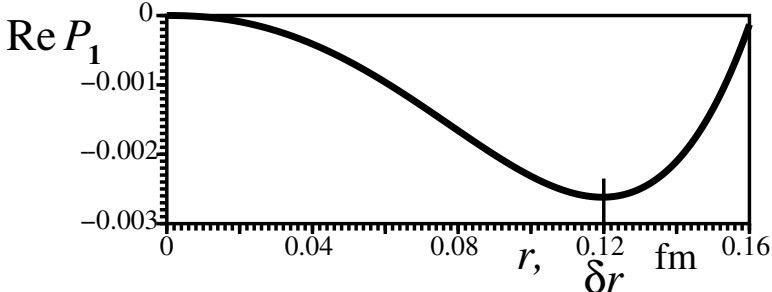

**Figure 10.** Real part of the pressure $P_1(r)$ in the region of $r < 0.16$ fm.

The region stretching below $R_0 = 0.161$ fm is that of large energy fluctuations—the smaller $r < R_1$, the larger they can be. It is a consequence of the uncertainty principle

$$\delta \vec{p} \cdot \delta \vec{r} \geq \frac{\hbar}{2}. \tag{126}$$

Let us evaluate the energy fluctuations at the point $\delta r = 0.12$ fm. The evaluations show that $E_p = \hbar \cdot c / 2e\delta r \approx 823$ MeV ($e$ is the electron charge). Difference of the value 823 MeV and the real proton mass expressed in energy units is $938 - 823 = 115$ MeV. This missing part is most likely accounted for by the remaining wave component that extends from 0.161 fm to about 0.84 fm.

On the other hand, electron scattering with a wavelength of $\lambda = 10^{-16}$ m shows that in the region below $R_0 = 0.16$ fm $= 1.6 \times 10^{-16}$ m point centers are provoking this scattering. They can be quark manifestations. It can be the new physical phenomenon mentioned above. The results of frontal collisions of high-energy particles on colliders show the creation of quark-antiquark pairs. It can be confirmation of the quark theory in addition to other experiments with hadron jets.

Note that in the core $r < R0$, the pressure accepts complex values. It may point to the existence of the quark-gluon plasma—qualitatively other existence of matter. $R_0$ is a confinement wall behind which a complex quark-gluon dynamics takes place. Below we consider some topological, string models of quarks on toruses. It turns that confinement is a native property of the topological invariants.

### 4.4.2. String Topological Models of Quarks

If we roll up the vortex tube, shown, for example, in Figure 2, in a ring and glue together its opposite ends, we obtain a helicoidal vortex ring [105]. The following set of equations

$$
\begin{aligned}
x &= (b + a\cos(\omega_0 t + \phi_0))\cos(\omega_1 t + \phi_1), \\
y &= (b + a\cos(\omega_0 t + \phi_0))\sin(\omega_1 t + \phi_1), \\
z &= a\sin(\omega_0 t + \phi_0)
\end{aligned}
\tag{127}
$$

specifies in the Cartesian coordinate system the positions of points on the vortex ring. Here $a$ is the radius of the tube; $b$ is the radius of the torus that is the distance from the origin to the tube axis. The frequency $\omega_0$ is that of spinning about the tube axis. While the frequency $\omega_1$ is that of rotation along the tube having the glued ends. The phase constants $\phi_0$ and $\phi_1$ can take arbitrary values between 0 and $2\pi$.

Note that the volume and the surface area of a torus have the following values

$$
V = 2\pi^2 ba^2, \qquad S = 4\pi^2 ba. \tag{128}
$$

Observe that both volume and its surface tend to zero regardless, whether the radius $a$ tends to zero or the radius $b$. In the first case, the torus degenerates to the one-dimensional ring. The second case represents an intriguing variant. It would seem that the radius $a$ is different from zero, but the surface of the sphere is zero. Most likely, this sphere has a double covering with oppositely oriented normals that annihilate at a meeting [78]. As a result, the volume and surface vanish.

Let us consider strings rolled on the torus. For the sake of simplicity, the torus has the radiuses $a = 1$ and $b = 2$. The frequencies $\omega_0$ and $\omega_1$ are signatures of the strings. The first frequency, for simplicity, is $\omega_0 = 1$. While, the second frequency is $\omega_1 = 1/3$ for the string shown in blue in Figure 11A and $\omega_1 = 2/3$ for the string shown in red in Figure 12A. Take attention that the second frequency is multiple of the subharmonic, where the divisor is 3.

Note that for strings having different frequencies $\omega_1$, we choose colors such as is adopted at the identification of quarks. Namely, the lower quark with a fractional charge $-e/3$ has the color green, and the two top quarks with a fractional one of $2e/3$ have colors red and blue.

Let $b$ tend to zero. At $b = 0$, the torus degenerates to a spindle sphere (the sphere with doubled covering). These cases are shown in Figures 11B and 12B. Such spheres possess punctured points on their poles. In these points, the string currents go onto the inner surface of the sphere either to the outer. This fact manifests itself by inverting the normal vector $\vec{n}_k$ ($k = 1, 2, \cdots, 6$) at passing through these punctured points. As a result, at calculating the surface area of the sphere, we get zero. It follows from the fact that the oppositely oriented normals cancel each other.

Let us trace the current moves along the string, starting from the top pole and ending with the same top pole. The revolution of the current is whole when the arrows become oriented in one direction. It happens after three passings through the lower pole. After every passing through either the lower pole or the top, one orientation of the normal vector $\vec{n}$ inverts. This means that the current flows either on the outer surface of the sphere or on the inner. The whole revolution is $3 \cdot 2\pi \to 3 \cdot 360^o = 1080^o$.

Figure 13 shows from the left a general view of three strings superposed on the doubled sphere with the radius $a = 1$. They form a perfect hexagonal triad. However, as follows from the uncertainty principle (126), this triad will blur. Nevertheless, a shared action of the uncertainty principle and the confinement of the color quarks within a finite volume leads to a self-sustained configuration. Here the Pauli exclusion principle provides that no two identical fermions can simultaneously occupy the same quantum state.

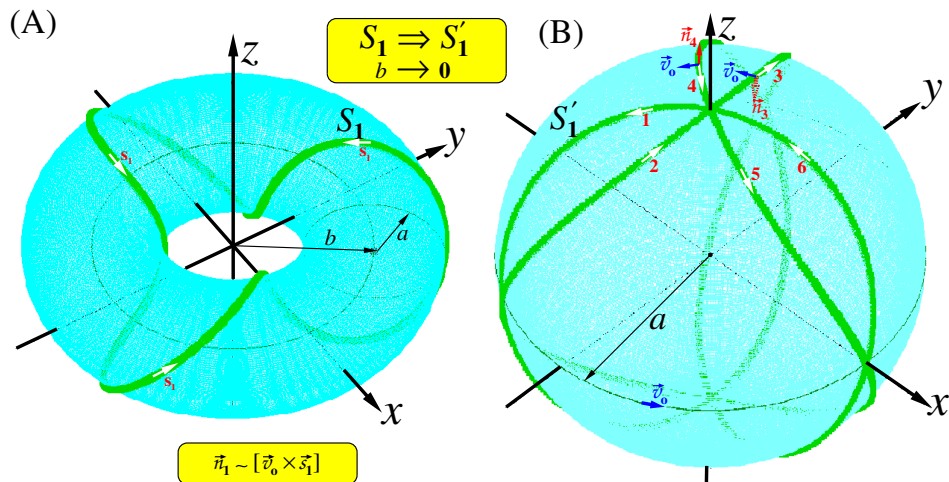

**Figure 11.** (**A**). Green string $S_1$ having parameters $\omega_0 = 1$, $\phi_0 = 0$, and $\omega_1 = 1/3$, $\phi_1 = \pi$ is drawn on surface of the torus having radiuses $a = 1$, $b = 2$. The current flowing along the string is drawn by white arrows and marked by red letters $s_1$; (**B**). Green string $S_1'$ drawn on the surface of the doubled sphere with radiuses $a = 1$, $b = 0$. The current flowing along this string is drawn by white arrows and marked by red numbers 1, 2, 3, 4, 5, 6. Blue arrows point rotation of the sphere about axis $z$ with the orbital velocity $\vec{v}_o$. The cross product of the velocity $\vec{v}_o$ by the current gives the normal to the sphere surface that is oriented either outside the surface (red arrow $\vec{n}_4$) or inside it (punctuated red arrow $\vec{n}_3$).

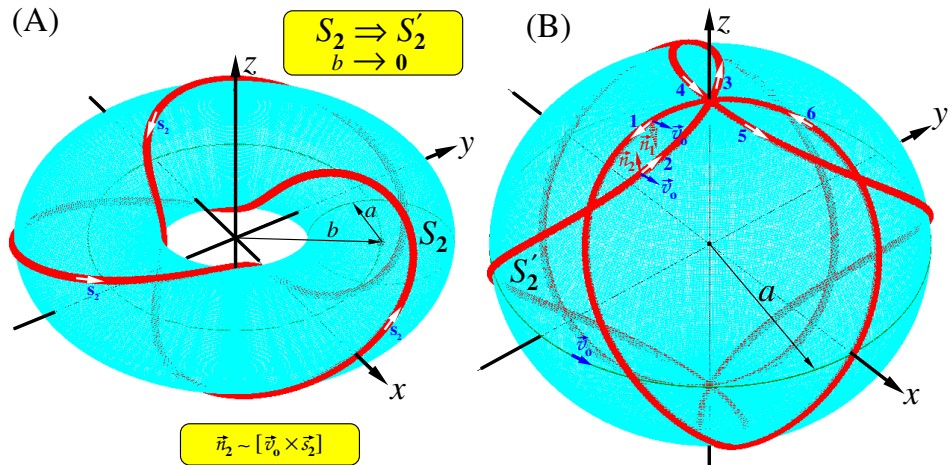

**Figure 12.** (**A**). Red string $S_2$ having parameters $\omega_0 = 1$, $\phi_0 = 0$, and $\omega_1 = 2/3$, $\phi_1 = \pi$ is drawn on surface of the torus having radiuses $a = 1$, $b = 2$. The current flowing along the string is drawn by white arrows and marked by blue letters $s_2$; (**B**). Red string $S_2'$ drawn on the surface of the doubled sphere with radiuses $a = 1$, $b = 0$. The current flowing along this string is drawn by white arrows and marked by blue numbers 1, 2, 3, 4, 5, 6. Blue arrows point rotation of the sphere about axis $z$ with the orbital velocity $\vec{v}_o$. The cross product of the velocity $\vec{v}_o$ by the current gives the normal to the sphere surface that is oriented either inside the surface (punctuated red arrow $\vec{n}_1$) or outside it (red arrow $\vec{n}_2$).

What happens if we apply the CPT transformation to these strings? First, the *P* (parity) transformation is equivalent to the inversion of the spatial axes. Observe that the positions of the red and blue strings do not change. Whereas the green string moves on the phase $\phi_0 = \pi$ from its initial position, see the right figure in Figure 13. As for the *C* (charge conjugation) transformation, it changes colors red to cyan, blue to yellow, green to pink, which is equivalent to the change of charges $2e/3 \rightarrow -2e/3$ and $-e/3 \rightarrow e/3$. The *T* (time reversal) transformation changes the direction of the currents along the strings. It means that a transformed object is the antiparticle.

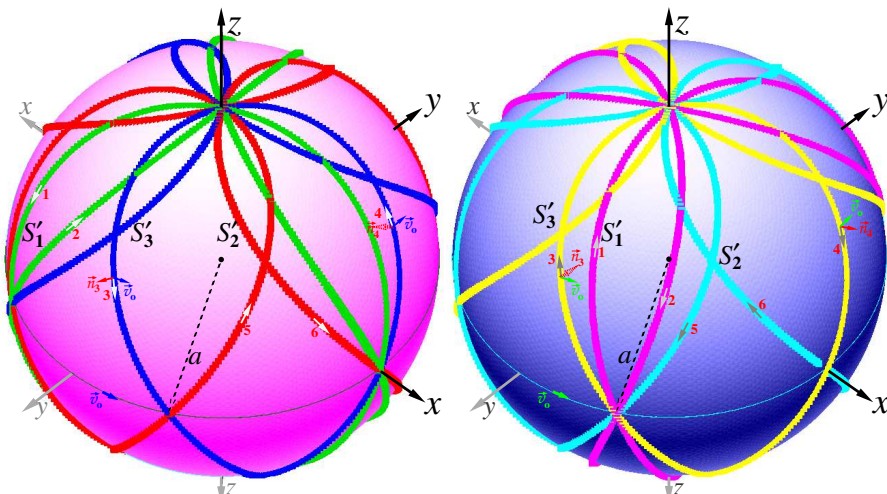

**Figure 13.** Three strings superimposed on the sphere of the radius $a = 1$ shown from the left are as follows: $S_1'$ (colored in green) has parameters $\omega_1 = 1/3$, $\phi_1 = 0$; $S_2'$ (colored in red) has parameters $\omega_1 = 2/3$, $\phi_1 = \pi$; $S_3'$ (colored in blue) has parameters $\omega_1 = 2/3$, $\phi_1 = 0$. The other parameters for all strings are $\omega_0 = 1$, $\phi_0 = 0$. White arrows, numbered by 1, 2, 3, 4, 5, 6, mark the current flowing along strings. Blue arrows point the direction of the orbital velocity $\vec{v}_o$. For the currents 3 and 4 orientation of the normals is shown—the normal $\vec{n}_3$ is oriented to outside (red arrow) and $\vec{n}_4$ inside (punctuated red arrow). The right figure represents the CPT transformation of ensemble of the left strings.

The CPT transformation by shifting the green string by the $\pi$ phase to a new position breaks the initial symmetry due to the shifting phase. There is no ideal merger of the initial object with the CPT transformed one. For that reason, they annihilate in a cascade.

Can long-lived pairs of baryons and antibaryons be formed? In our case, this question concerns the possibility of the formation of long-lived quark and antiquark triplet unifications, the existence of so-called hexaquarks [106]. It is a large family of hypothetical particles. Each family contains six quarks or antiquarks of any flavor. Hexaquart may represent two related to each other baryon (dibaryon) [107]. According to forecasts, such a formation of dibaryons can be stable quite. Hexaquarks d*(2380) consisting of six light quarks; 3*u*-quarks and 3*d*-quarks [108] can form a substance known as Bose–Einstein condensate (BEC) due to the prevailing low temperatures in the Universe. Under some conditions, BEC containing hexaquarks with the captured electrons and positrons can behave as dark matter.

Ethan Siegel writes in his digest [109]: *"even if it is possible to create a d* condensate as the authors propose, it cannot survive the intense radiation of the early Universe. Once they are blasted apart, there is no way to create more d* particles capable of forming a Bose–Einstein condensate, as the conditions that admit their creation will have passed. It is a clever idea, but we do not need to wait for colliders to rule it out. The early Universe as we understand it is already enough to crush the idea that d* hexaquarks can make up our Universe's dark matter."* So far, no one knows what dark matter is. However, the mass of indirect data shows that it exists [6,14–16,110–112]. All also agree that this matter manifests itself through the Bose–Einstein condensate [26,28,30,34,35,113,114].

Some scientists believe [34,56,115,116] that possible candidates on the dark matter carrier could be Weyl, Dirac, and Majorana spinor fields. The paired Majorana fermions [99] can form self-sustained spinor fields representing by themselves a matter and antimatter. The most stable baryons and leptons are protons and electrons. Stable particles are also their antipodes, i.e., antiproton and positron. Their lifetime is infinite. Neutrons and antineutrons have no infinite lifetime. However, they can live infinitely long in cooperation with protons and antiprotons. One believes that only these material particles (protons, neutrons, electrons) survived until our epoch. As for antimaterial particles, their absence confuses the scientific community. We can assume that most of the antimatter form a self-sustaining condensate

with the matter. This condensate is the Bose–Einstein condensate at low cosmic temperatures. At this stage, the atoms move from their normal state to a completely stationary, minimal possible quantum state. Due to the Meissner effect, this condensate becomes invisible, except for the gravitational influence.

The simplest combination is the proton-antiproton pair. It contains *uud* and $\bar{d}\bar{u}\bar{u}$ quarks. Why is a negatively charged electron not absorbed by a positively charged proton? It is because that the de Broglie wavelength of the electron imposes a ban on this absorption (unless the energy of the electron exceeds a certain threshold). The de Broglie wavelength keeps the electron in a given quantum orbit, called the Bohr orbit. The same situation can be possible for positively charged proton and negatively charged antiproton, keeping them in a quantum orbit.

### 5. Nonelectromagnetic Forces of Rotating Masses in Vacuum

Observations show that the amount of dark matter exceeds five times the amount of ordinary matter. That is, dark matter is ubiquitous. It permeates all the surrounding space and everywhere its presence exerts indirect action. Note that the dark matter can only manifest itself through gravitational and torsion forces but not electromagnetically.

In this vein, the experiment performed by V. Samokhvalov [117] has particularly significant. The essence of which lies in the fact that a lower drive disk, Figure 14, is driven in rotation in a deep vacuum. After a while, it involves in this rotation the upper, slave, disk. One should emphasize that there are no mechanical connections between the disks. Samokhvalov also made sure that all the details in the experiment were dielectric and nonferromagnetic.

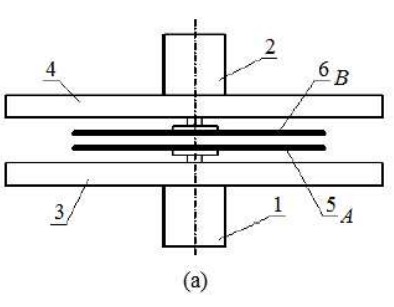 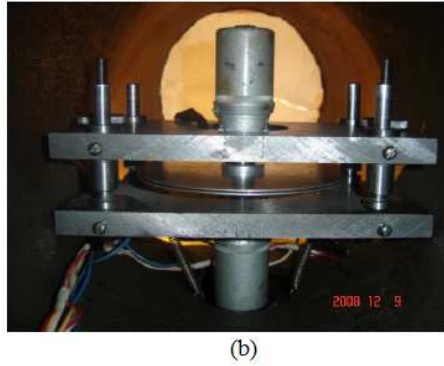

**Figure 14.** Samokhvalov's device [118] for studying the mass-dynamical effect: (**a**) schematic diagram: 1 and 2 are axes with electric motors mounted on them, 3 and 4—steel boards with electromagnetic brakes fixed on them, 5 and 6—nonferromagnetic disks with a radius $R = 8.25$ cm rigidly fixed to the flanges of the rotors of electric motors (the gap between the disks is from 1 to 6 mm or more, the maximum frequency of disks spin of the order of 100–120 1/s); (**b**) general view of the device.

One can note the following observable facts [119]: (i) the experiment states that the force acting in a vacuum from the driving disk No. 5 causes spinning the slave disk No. 6, mechanically not connected with the former; (ii) it shows that the rotation forced of the initially stationary slave disk No. 6 is the consequence of its noncontact interaction in a vacuum with the rotating disk No. 5. In the presence of air in the chamber, the forced rotation of disk No. 6 does not occur; (iii) it is established that the repulsion forces from the side of the driving disk No. 5 act on the slave disk No. 6 (like the repulsive Casimir effect [120–123]). The effect of disk's repulsion is manifested only under the condition of imbalance of the driving disk No. 5; (iv) the experiments have shown that with a sufficiently high degree of disk balancing and the absence of vibration, the forced rotation of the slave disk No. 6 was not excited. This experiment attracts the attention of many scientists [69,117,119,124–127]. The consensus is that special gravitomagnetic forces with some modifications are involved [128].

Here we follow the ideas stated in the theory of the superfluid $He^3$ [30,34,94,129,130]. In particular, Lounasmaa and Thuneberg at their article [94] have evaluated number $N$ of vortices in a rotating container, with the radius $R$, arising as the angular velocity $\Omega$ varies:

$$N = 2\pi R^2 \cdot \frac{\Omega}{\kappa_0} = \pi R^2 \cdot \frac{\Omega}{\Phi_m} = S \cdot \frac{2m\Omega}{h}. \tag{129}$$

Here $\Phi_m$ is the quantum of the vorticity flux (100), $\kappa_0$ is the circulation quantum named by Lounasmaa and Thuneberg in [94], and $S = \pi R^2$ is the area of a disk, Figure 15.

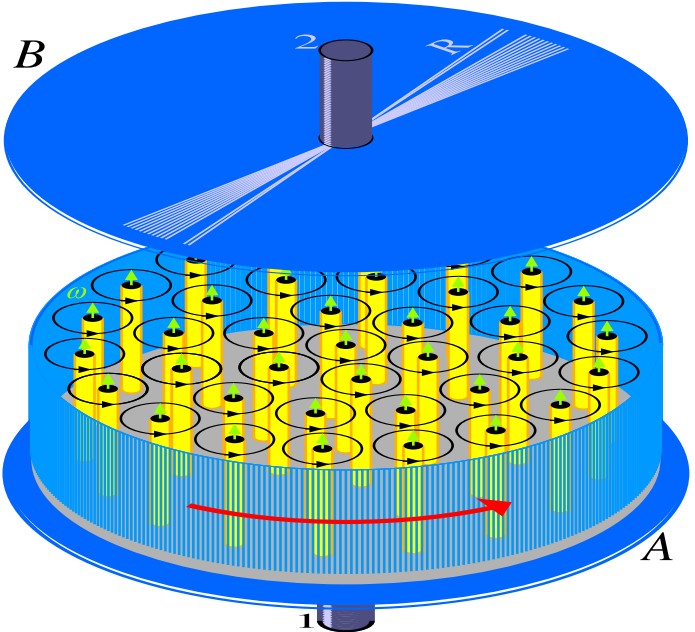

**Figure 15.** The rotating disk A involves the driven disk B to rotation by grabbing it through the vortex filaments (drawn by yellow cylinders) growing from the bottom drive disk [131]. Green arrows represent the vorticity $\vec{\omega}$ black circles with arrows depict vortex rotation of the yellow cylinders. The both have radius R impaled on unbound axes 1, and 2.

Substituting in this formula the disk radius, $R = 0.0825$ m, and its angular velocity of rotation, $\Omega$, equal about $100 \, \text{s}^{-1}$ [118], and also the twice mass of proton $m = 2 \cdot 1.67 \times 10^{-27} = 3.34 \times 10^{-27}$ kg (sum of proton and antiproton masses) we find that the number of the vortices can reach $10^7$ and over.

*Neutron Interference Experiment Project*

Note that each vortex filament is weak enough, but millions of vortices can be able to anchor and drag the upper disk as soon as they reach its surface [130]. It is of interest to set such an experiment with rotating massive disks on a neutron beam for studying the formation of such vortex filaments. The beam splits into two, each of which passes on different sides of the rotating masses, Figure 16. At the opposite end, they are brought together again and fed to a detecting device. Here one recognizes the classic Bohm–Aharonov experiment [132] when the result of interference on the detector of two united beams can indicate the presence or absence of rotation of the second mass because of growing the vortex filaments.

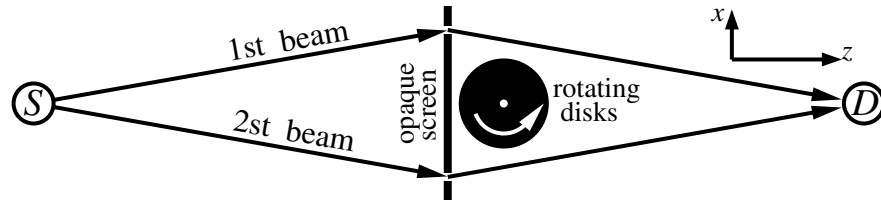

**Figure 16.** Scheme of the Aharonov–Bohm experiment for the two massive rotating disks placed in a deep vacuum. Two neutron beams from source *S* pass through two slits in an opaque screen and converge on detector *D* where the interference fringes form. The rotating disks are between the neutron beams in a screen shadow.

Let us evaluate the probability amplitude of particle registration by the detector *D* at its radiation by the source *S*, Figure 16 by using the Feynman path integral technique [133]. A full probability amplitude is equal to the integral convolution of two kernels, each describing movement of the free particle from the source *S* to localizations of the slits on the opaque screen and from the slits to the detector position [134]:

$$\psi(x_1, x_0, x_s) = \int_{-b}^{b} K(x_1, T + \tau; x_0 + \xi, T) K(x_0 + \xi, T; x_s, 0) d\xi \qquad (130)$$

Here the kernel reads

$$K(x_a, t_a; x_b, t_b) = \left[ \frac{2\pi \mathbf{i}\hbar(t_b - t_a)}{m} \right]^{-1/2} \exp\left\{ \frac{\mathbf{i}m(x_a - x_b)^2}{2\hbar(t_a - t_b)} \right\} \qquad (131)$$

After a series of computations [134,135], we get the wave function having the following view:

$$|\psi(z, x)\rangle = \frac{1}{N\sqrt{1 + \mathbf{i}\dfrac{z\lambda_{dB}}{2\pi b^2}}} \sum_{n=0}^{N-1} \exp\left( -\frac{\left( x - \left( n - \dfrac{N-1}{2} \right) d \right)^2}{2b^2 \left( 1 + \mathbf{i}\dfrac{z\lambda_{dB}}{2\pi b^2} \right)} \right). \qquad (132)$$

This function in the limit of the number of slits, *N*, and the distance between them, *d*, the both tending to infinity reproduces a fractal pattern named in literature Talbot carpet [136,137]. The other parameters in this function are de Broglie wavelength $\lambda_{dB}$ and the width of slits *b*.

The Talbot carpet, Figure 17, certifies the truth of the wave function (132). Further, we take this function as that emitted from the two-slits device

$$|\Psi(z, x)\rangle = \overbrace{\frac{1}{\sqrt{1 + \mathbf{i}\dfrac{\lambda_{dB} \cdot z}{2\pi b^2}}}}^{(a)} \times \left( \exp\left( -\frac{\left( x + \dfrac{1}{2}d \right)^2}{2b^2 \left( 1 + \mathbf{i}\dfrac{\lambda_1 \cdot z}{2\pi b^2} \right)} \right) + \exp\left( -\frac{\left( x - \dfrac{1}{2}d \right)^2}{2b^2 \left( 1 + \mathbf{i}\dfrac{\lambda_2 \cdot z}{2\pi b^2} \right)} \right) \right). \qquad (133)$$

and we choose only parameters corresponding to the experiment drawn in Figure 16. The Aharonov–Bohm neutron interference experiment is in a very near zone, i.e., the rotating disks placed between slits in the shadow zone do not illuminate by the beams. If we take the cool neutrons (the de Broglie wavelength $\lambda_{dB}$ is about 0.7 nm) and the distance between slits let be about 40 cm, than the very near zone is $z \approx 20$ m $\ll z_T = d^2/\lambda_{dB} \approx 10^8$ m, see Figure 18.

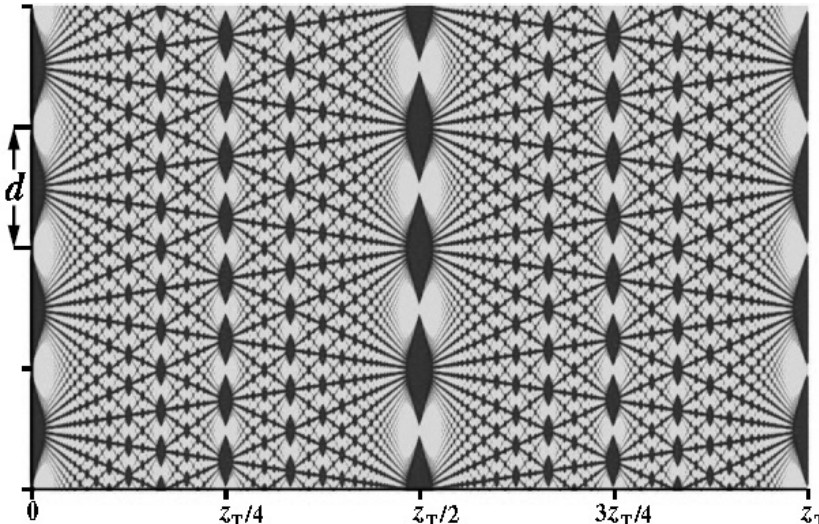

**Figure 17.** The density distribution $p(z, x) = \langle \psi(z, x) | \psi(z, x) \rangle$ taken from central part of the interference grating shows the Talbot carpet at choosing $N = 64 \gg 1$ of slits. The central slits are $n = 30, 31, 32, 33, 34$. Here $z_\mathrm{T} = d^2 / \lambda_\mathrm{dB}$ is a unit of length named the Talbot length.

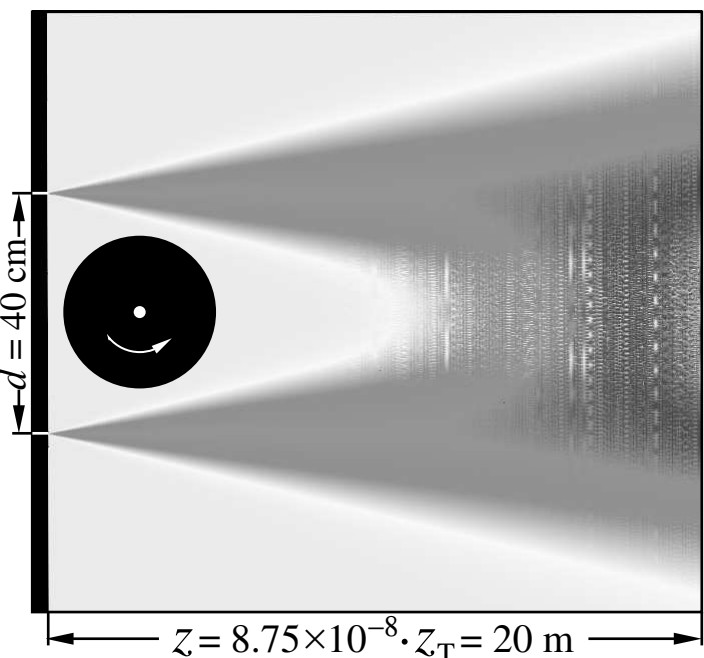

**Figure 18.** The density distribution $p(z, x) = \langle \psi(z, x) | \psi(z, x) \rangle$ calculated from the wave function (133) with omitting the amplitude factor. The rotating disks are placed in the shadow zone.

The perturbed de Broglie wavelengths $\lambda_{1,2} = \lambda_\mathrm{dB}(1 \mp \delta)$, $\delta \ll 1$, are represented only in exponents of Equation (133). Note that $\delta$ introduced in the amplitude multiplier (by embraced by a brace (a)) does not lead to variations of the interference fringes. For that reason, this multiplier has only the pure de Broglie wavelength. The appearance of the term $\delta$ is due to the presence of orbital momentum $m\vec{v}_o$ in the Schrödinger equation (90). From here we conclude that the wavelengths $\lambda_1$ and $\lambda_2$ follow from the formula:

$$\lambda_{1,2} = \frac{h}{m_n v_S \pm m v_o} \approx \lambda_\mathrm{dB}(1 \mp \delta). \tag{134}$$

Here $\lambda_{\text{dB}} = h/(m_n v_S)$ is de Broglie wavelength, $v_S$ is the irrotational speed coming from the gradient of the action function $S$, and

$$\delta = \frac{m v_o}{m_n v_S} \ll 1. \tag{135}$$

The term $m \vec{v}_o$ in Equation (90) is analogous to the vector potential $\vec{A}$ with loaded by the factor $2e/c$.

If $\delta$ is not zero what leads to a splitting the de Broglie wavelength, see Equation (134), then the density distribution $p(x,z) = \langle \psi(x,z)|\psi(x,z)\rangle$ shows different values on detectors located at mirror positions with respect to the central line. Figure 19 shows an example of the interference fringes arising on a distance $z = 20$ m far from the opaque screen with two slits made in it. Red lines mark positions of detectors spaced conditionally at a distance of $\pm 1$ cm from the central line. One can see, as soon as the split of the de Broglie wavelength arises, the detectors show a difference in indications.

At choosing the $\delta$ ranging from $10^{-6}$ to $10^{-7}$ the difference $\lambda_\delta = \lambda_2 - \lambda_1 = 2\lambda_{\text{dB}} \cdot \delta$ ranges between 1.4 fm and 0.14 fm. This difference overlaps the proton radius and deeps to the confinement length of quarks, Figures 8–10. One can evaluate energy $E = \hbar \cdot c/(e\lambda_\delta)$. It falls in the range from about 140 MeV to 1400 MeV. Observe that the interference of two coherent rays is very sensitive to tiny scales. Therefore, it can serve, for studying, a delicate quark–gluon behavior within baryon matter.

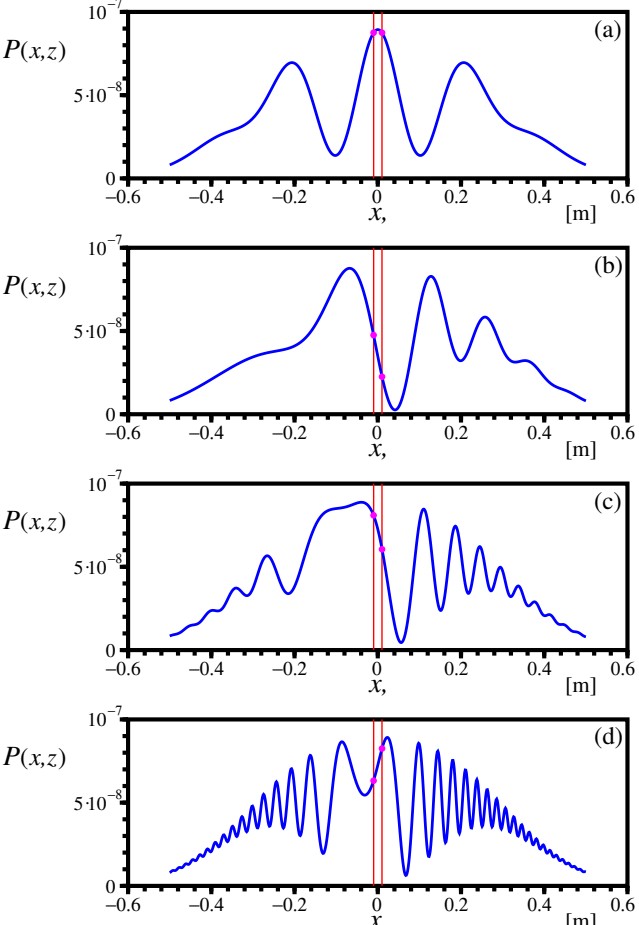

**Figure 19.** Interference fringes (blue curves) written at the distance of $z = 20$ m from the opaque screen, see Figure 16 at different values of $\delta$: (**a**) $\delta = 0$; (**b**) $\delta = 10^{-7}$; (**c**) $\delta = 4 \cdot 10^{-7}$; (**d**) $\delta = 10^{-6}$. Red lines show positions of detectors at $x_D = \pm 0.01$ m; Pink dots on intersection of blue curves and red lines are detector registered values.

## 6. Conclusions

Quaternion algebra makes it possible to describe motions of matter on a four-dimensional space-time naturally. The gravitomagnetic equations describing the states of both the gravity-torsion field (the torsion field of massive objects) and the electromagnetic field stems from such an approach.

The gravitomagnetic equations give rise to a series of equations, among which the modified Navier–Stokes equation is perhaps one of the most intriguing. This equation describes the velocity field of a superfluid quantum medium that everywhere densely fills 4D space-time. The property of this medium is that its viscosity is zero on average in time. On the other hand, the variance of the viscosity coefficient is nonzero. It can mean that there exists a permanent exchange of mass and energy between the superfluid component of the medium (a true Bose–Einstein condensate (BEC)) and the normal one.

The superfluid and normal components are mutually complementary components of the medium, the existence of which directly follows from the Ginzburg–Landau theory of superfluidity [87].

The normal component is that where fluctuations undergo dissipation. Fick's laws are valid in this component. These laws applied to the modified pressure gradient reveal a quantum potential inherent to the modified Navier–Stokes equation (really, the modified pressure gradient maps to the quantum potential gradient). Besides, the continuity equation describes the distribution of the mass density of this medium. One can conclude that we deal with the superfluid quantum ether where the BEC represents the dark matter.

Together, the two equations above-mentioned give rise to the Schrodinger equation as soon as we define the complex-valued wavefunction. This function includes information about the velocities field (described by the Navier–Stokes equation) and the mass density distribution (relating to the continuity equation). From here it follows, that the wave function is a real physical object that describes the state of a superfluid quantum medium everywhere densely filling 4D space-time.

As noted above, the quantum potential acts on the normal component. For this reason, its effect cannot extend further than on the diffusion length—the square root of the diffusion coefficient multiplied by the average diffusion time. What is the essence of "spooky action on a distance"? It occurs due to the transfer of quantum information to the superfluid component, which causes a nondissipative action over gigantic distances.

As follows from the Ginzburg–Landau theory of superfluidity, the superfluid component is an ideal BEC where the Bose particle is pair of coupled Fermi particles with half-integer spins. As a result, a paired particle-antiparticle gives a single quantum object having an integer spin. Following Occam's razor principle, such particles and antiparticles most possible are protons and antiprotons. It is quite natural that the normal component for such a superfluid component can be hydrogen and helium, widely distributed in the Universe.

At low temperatures, both the $He^3$ pairs and the proton-antiproton pairs can form BEC. When this condensate rotates, vortex filaments appear that penetrate this superfluid medium along the axis of rotation. Vortex filaments are represented by revolution around the center of mass of pairs of helium atoms in the case of superfluid helium or paired proton-antiproton in the case of superfluid ether. The aim of the Aharonov–Bohm interference experiment on cold neutrons is to evaluate the mobility of these filaments.

Interferometry provides a perfect opportunity to study phenomena at much smaller scales than the de Broglie wavelength of the quantum object of interest. It is because interferometers measure the difference in the wavelengths coming to the detectors along two different paths. We have considered the occurrence of vortex filaments between rotating massive disks in a deep vacuum [118,124]. It is a phenomenon that is similar to the appearance of vortex filaments in a rotating cylinder filled with superfluid $He^3$, described by Lounasmaa and by Thunberg in the article [94]. In both cases, the question arises about the organization of vortex filaments due to the rotation of two baryon particles relative to

each other. In the latter case, they are He$^3$ atoms. As for the first case, here we are dealing with the rotation of a particle-antiparticle pair, more precisely, a paired proton-antiproton.

Note that baryonic particles—He$^3$ pairs and proton-antiproton pairs—consist of quarks that interact with each other in a complicated way. There is reason to believe that these complex interactions give rise to a new state of quarks, called hexaquarks [107,108]. For this reason, the cold neutron interference experiment, due to its high sensitivity, can shed light on such a quark conglomerate.

Can strange quarks [106,109] be present in the superfluid quantum ether, or does it consist of ordinary up and down quarks? The question concerns the essence of dark matter. The author adheres to the idea that the dark matter that fills the Universe in the modern era consists mainly of ordinary hexaquarks, i.e., combinations of quarks *uud* and *d̄ūū*. Other quark combinations, such as strange and charming quarks and higher quarks (upper and lower), most likely burned out in the early stages of the Universe's evolution.

Finally, one can note that the superfluid BEC may be a source of plasmoids, ball lightning, that sometimes are observed in the fault zones of geological plates (in anomalous zones). Perhaps ancient civilizations such as the Egyptian one knew about this and could use such sources as perfect energy employing pyramids [138].

**Funding:** This research received no external funding.

**Acknowledgments:** The author has to thank Mike Cavedon, Marko Fedi, and many colleagues of the seminar in my institute for fruitful discussions. The author also thanks Reviewers for productive and positive critical remarks.

**Conflicts of Interest:** The author declares no conflict of interest.

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
