# Peer review of "Quaternion Algebra on 4D Superfluid Quantum Space-Time: Can Dark Matter Be a Manifestation of the Superfluid Ether?"

_universe, doi:10.3390/universe7020032_

Round 1

Reviewer 1 Report

I suggest checking the copyright for the use of the images in the text. If these are granted, I recommend the paper for publication in the present form.

Author Response

I tried to take into account the comments of Reviewer 1 to the manuscript universe-1005293. Below are the items that have been rewritten and corrected.

Response 1: I have rewritten the introduction. It begins from four-dimensional pseudo-Euclidean Minkowski space-time. And from the determination of the energy-momentum tensor written customary in the relativistic hydrodynamics. Further, it is mentioned Penrose's new type of algebra for Minkowski space-time. This algebra opens the presence of the torsion in space-time.  That consistent apparently with the local torsion of the Einstein-Cartan-Sciama-Kibble.  One can see the quaternion algebra on 4D space-time represents a lateral branch of Penrose's twister program.

Response 2: I have given an explanation of the energy and momentum densities depending on the mass density by adding phrases (a) "Note that these formulas we write under the assumption of relativistic transport fluxes in an ideal fluid." after Eq.(18), and (b) "The function $\rho$ plays an important role in the quantum realm. It determines the probability density of detecting particles within the volume $\Delta V$ at the moment $t$. In particular, this function enters to the definition of the quantum potential~(\ref{eq=5})." after Eq.(19).

Response 3: I have given interpretations of the differential operators D and D^T generating the gravitomagnetic equations.

Response 4: I have emphasized differences between the customarily defined energy-momentum tensor (1) and that of (22) considered in this article.

Response 5: The gravitomagnetic field tensor is defined in Eq.(25).

Response 6: As for the transposed derivative operator, one can begin from this operator, and then the following operator should be the derivative operator. Multiplication of these two operators leads to the d'Alambertian, as shown in Eq.(48).

Response 7: I deleted the computation chain leading to the formula of the black body radiation.

Response 8: Instead of the Schrodinger equation, I have derived the Schrodinger-Pauli equation.  The latter describes the behavior of the spin of a particle in a magnetic field. As a result, at derivation of the Ginzburg-Landau equation, the magnetic field written in the quaternion basis arises in this equation. Since the Meissner effect rejects the presence of a magnetic field in the superfluid component, the magnetic terms in the Ginzburg-Landau equations collapse.

Reviewer 2 Report

Starting with a quaternion formalism to describe our 4D space-time, the author is able to gravitomagnetic equations which have a strong similarity to the Maxwell-equations. From these equations, he is able to derive modified Navier-Stokes equations which lead support to a BOSE-Einstein condensate approach to dark matter (the author refines this concept by associated dark matter to hexa-quarks later in the paper). He is also able to derive a vorticity equation and the Schroedinder and Dirac equations from his general approach.

The quaternionic approach to a description of space-time and the consequences outlined in this paper are quite impressive. The quaternion formalism has a lot of power not enough appreciated in the literature. The encyclopedic approach of this paper very much support the use of quaternion in describing physics and, as such, deserve publication. The list of references is quite complete and will be of great interest for researchers interested in learning more about the quaternionic approach to describing physics.

The paper needs some clarification in the presentation:

In Eq.(1) and right underneath, the author assumed that the speed of light c is equal to 1. Later on in the text, c appears explicitly in the equations. There should be some consistency in the presentation.

There are many typographical errors:

For instance,

Below Eq.(9), it should be gravitomagnetic, not gravitomagmetic.

Below Eq.(13), it should be SU(2), not S(2).

In the conclusions, “paisr”should be “pairs”

The author should run a spell-check on the entire article.

CMB (Cosmic Microwave Background) should be spelled out the first time it appears on page 3 of the paper.

Midway through page 3, it should read” While studying the rotation curves of galaxies, Vera Rubin found…”

Author Response

In Eq.(1) and right underneath, the author assumed that the speed of light c is equal to 1. Later on in the text, c appears explicitly in the equations. There should be some consistency in the presentation.

Response 1: I have rewritten Eq.(1) and part of the text with presenting the explicit speed of light c

Below Eq.(9), it should be gravitomagnetic, not gravitomagmetic

Response 2: Thank you. I corrected this word.

Below Eq.(13), it should be SU(2), not S(2)

Response 3: Thank you. I corrected S(2) to SU(2).

In the conclusions, “paisr”should be “pairs”

Response 4: Thank you. I corrected this word. 

CMB (Cosmic Microwave Background) should be spelled out the first time it appears on page 3 of the paper.

Response 5: Yes, I did it.

Midway through page 3, it should read” While studying the rotation curves of galaxies, Vera Rubin found…”

Response 6: Thank you, yes I did it.

This manuscript is a resubmission of an earlier submission. The following is a list of the peer review reports and author responses from that submission.

Round 1

Reviewer 1 Report

The paper does not meet the publishing standards as it is and I recommend a major revision. More comments can be found in the attached file.

Reviewer 2 Report

The subject of this paper contains the word ¨dark matter¨ , however nothing that is done in the paper itself shows that the constructions the author are in anyway related to Dark Matter, if fact the constructions have non trivial electromagnetic properties, so where is the dark aspect?.

Independently of this the paper the paper discusses diverse subjects, like ¨String topological models of quarks ¨, ¨Majorana fermions: proton-antiproton dancing ensemble¨, etc., that may be all very interesting, although I think also very unlikely since I do not see much coherence in the paper.

At this point I would not recommend acceptance of this paper